# 'Fearful-place' coding in the amygdala-hippocampal network

Mi-Seon Kong[1,2], Eun Joo Kim[1], Sanggeon Park[3,4], Larry S Zweifel[2,5], Yeowool Huh[4,6], Jeiwon Cho[3]*, Jeansok J Kim[1]*

[1]Department of Psychology, University of Washington, Seattle, United States; [2]Department of Psychiatry and Behavioral Sciences, University of Washington, Seattle, United States; [3]Department of Brain and Cognitive Sciences, Scranton College, Ewha Womans University, Seoul, Republic of Korea; [4]Institute for Bio-Medical Convergence, International St. Mary's Hospital, Catholic Kwandong University, Incheon, Republic of Korea; [5]Department of Pharmacology, University of Washington, Seattle, United States; [6]Department of Medical Science, College of Medicine, Catholic Kwandong University, Gangneung, Republic of Korea

**Abstract** Animals seeking survival needs must be able to assess different locations of threats in their habitat. However, the neural integration of spatial and risk information essential for guiding goal-directed behavior remains poorly understood. Thus, we investigated simultaneous activities of fear-responsive basal amygdala (BA) and place-responsive dorsal hippocampus (dHPC) neurons as rats left the safe nest to search for food in an exposed space and encountered a simulated 'predator.' In this realistic situation, BA cells increased their firing rates and dHPC place cells decreased their spatial stability near the threat. Importantly, only those dHPC cells synchronized with the predator-responsive BA cells remapped significantly as a function of escalating risk location. Moreover, optogenetic stimulation of BA neurons was sufficient to cause spatial avoidance behavior and disrupt place fields. These results suggest a dynamic interaction of BA's fear signalling cells and dHPC's spatial coding cells as animals traverse safe-danger areas of their environment.

*For correspondence:
jelectro21@ewha.ac.kr (JC);
jeansokk@u.washington.edu
(JJK)

Competing interest: The authors declare that no competing interests exist.

## Introduction

Biological fitness requires that all organisms foraging for resources, such as food, water, and shelter, be able to discern and respond appropriately to varied landscapes of danger (*Bolles, 1970*; *Lima and Dill, 1990*; *Pellman and Kim, 2016*). Consistent with this view, animal and human studies have found that distinct anti-predatory behaviors and neural circuits are engaged in distal vs. proximal threats (*Fanselow and Lester, 1988*; *Mobbs et al., 2007*). The basic fear system shared across species then likely evolved to guide and shape goal-directed (or purposive) behaviors in risky environments (*Brown et al., 1999*; *LeDoux, 2012*; *Maren and Fanselow, 1996*; *Ohman and Mineka, 2001*). To date, however, most neurobiological fear research has focused on the acquisition and extinction mechanisms of Pavlovian fear response magnitudes (*Fanselow and LeDoux, 1999*; *Maren and Quirk, 2004*), while overlooking how the brain responds to spatial dynamics of threats in nature.

Initial studies that explored the spatial component of danger continued to use basic conditioning paradigms and found that a conditioned freezing response was associated with remapping of dorsal hippocampal cornu ammonis 1 (CA1) place fields in rats randomly searching for food pellets in an experimental chamber where they previously received painful periorbital shocks (contextual fear) or in a control chamber when presented with conditioned white-noise pip stimulus (auditory fear; *Moita et al., 2003*; *Moita et al., 2004*). A subsequent study, simulating a scenario of a hunger-motivated prey leaving its nest to look for food and encountering a predator, revealed that fear reflexively

elicited by a looming robotic predator altered place cell activities in rats foraging for food in a large arena as a function of proximity to danger; an effect that was abolished with lesions to the amygdala (*Kim et al., 2015*). These studies then suggest that both acquired and innate fear can alter spatial representation in the hippocampus, irrespective of whether animals are displaying immobility (*Moita et al., 2003*; *Moita et al., 2004*) or rapid escape (*Kim et al., 2015*) fear responses. More recently, fear-induced remapping of CA1 place cell was also confirmed using miniscope calcium imaging and inhibitory avoidance task in mice (*Schuette et al., 2020*). However, since only place cell firing characteristics were examined in isolation in the aforementioned studies (*Kim et al., 2015*; *Moita et al., 2003*; *Moita et al., 2004*; *Schuette et al., 2020*), how the dimensions of fear and space interact in real time at the coding level as animals navigate risky environments remain unknown. To address this, the present study recorded, for the first time, neural activities simultaneously in the basal amygdala (BA) and dorsal hippocampus (dHPC), structures implicated in fear and spatial behavior, respectively, in rats using an ecologically relevant 'approach food-avoid predator' paradigm (*Choi and Kim, 2010*; *Kim et al., 2018*). We then applied optogenetics to determine the functional relationship between the brain's fear and spatial systems.

## Results

### Foraging behavior under a predatory risk

Rats (n = 4) implanted with micro-drive arrays in both BA and dHPC (*Figure 1—figure supplement 1A*) were trained to leave the nest area to procure a 0.5 g food pellet in a tapered foraging arena that enabled adequate visit maps for reliable place-responsive dHPC cell analyses and recurrent predatory interaction for consistent fear-responsive BA cell analyses (*Figure 1A*). Tetrodes were gradually advanced (<160 µm per day) until complex spike cells were encountered, which were identified on the basis of electroencephalogram signals and single-unit spike patterns. Upon encountering stable spiking cells, the hunger-motivated animals underwent successive 'pre-robot,' 'robot,' and 'post-robot' recording sessions (8–10 pellet attempts/session; *Figure 1A*). During the pre-robot session, all animals promptly exited the nest, obtained the pellet, and returned with it to the nest for consumption (100% success; *Figure 1B*). During the robot session, each time the animals approached the pellet, the looming robot caused them to flee to the nest. Specifically, rats exhibited significantly increased outbound foraging latency due to pauses in approaching the pellet (pre-robot, 2.73 ± 0.29 s; robot, 5.53 ± 0.57 s; post-robot, 2.97 ± 0.18 s; *Figure 1B*) and decreased ability to secure the pellet (3.6% success; *Figure 1B*). Representative trajectories during each session showed that the rat traveled more distance during the robot session, indicating multiple failed attempts to retrieve the pellet because of the predatory threats (*Figure 1A*). Once the robot was removed (the post-robot session), all rats subsequently reverted to the pre-robot foraging success rate (100%). Because the looming robot prevented the animals from reaching the pellet location, subsequent analyses excluded those dHPC cells that had place fields beyond the foraging limit (where the animal did not visit during the robot session) to equate the nest-to-foraging distance throughout the sessions (*Figure 1A*, yellow-tinted dotted line).

### Spike synchrony between BA and dHPC units during the predatory encounter

To investigate whether and how fear coding BA cells and spatial coding dHPC cells interact during an 'approach food-avoid predator' conflict situation, we performed simultaneous single-unit recordings from these two brain regions during risky foraging behavior paradigm (*Figure 1C*, *Figure 1— figure supplement 1A and B*). To directly assess spike synchrony while rats attempted to procure a pellet in a fearful situation, we generated cross-correlograms (CCs) with BA cells as the reference with four different time epochs: (i) pre-pellet, 2.5 s epoch before pellet procurement during the pre-robot session; (ii) post-pellet, 2.5 s epoch after the pellet procurement during the pre-robot session; (iii) pre-surge, 2.5 s epoch before the robot activation during the robot session; and (iv) post-surge, 2.5 s epoch after the robot activation during the robot session (*Figure 1D*). The epoch size (2.5 s) for assessing BA-dHPC correlational firing was chosen based on the mean outbound foraging time during the pre-robot session, which reflects the time when rats were supposedly in the foraging area (*Figure 1B*). The raw CCs were corrected by 'shift-predictor,' where 100 times of trial shuffles were

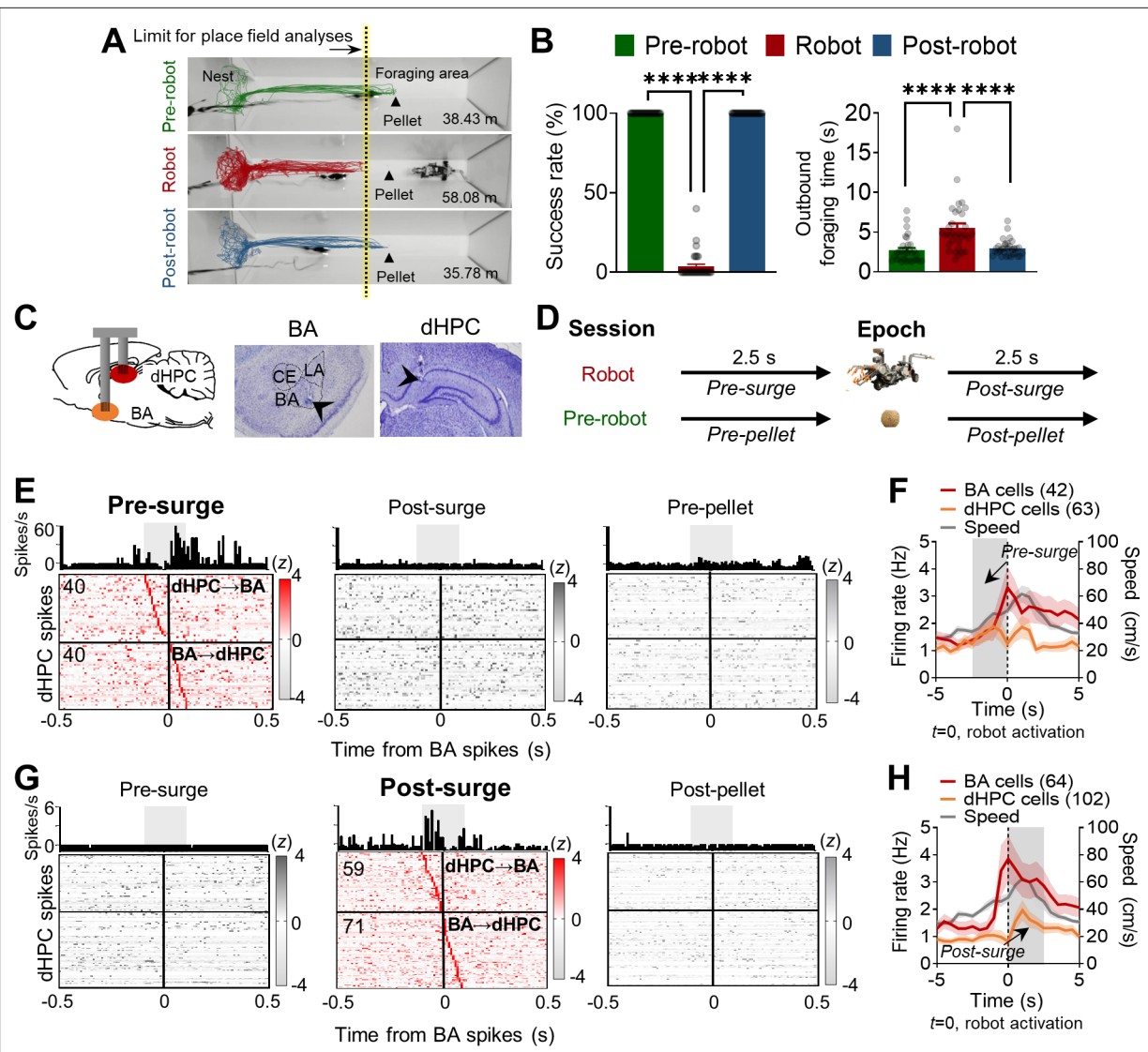

**Figure 1.** Simultaneous basal amygdala (B) and dorsal hippocampus (dHPC) recordings in foraging rats facing a predatory threat. (**A**) Overlaid images of the foraging apparatus and representative trajectories of a rat during the pre-robot, robot, and post-robot sessions. The number below each apparatus indicates the total distance traveled from the representative trajectory data. Contrary to 100% successful trials during the pre-robot and post-robot sessions, this rat made 14 attempts to procure the pellet during the robot session but failed each time. The limits of place field analyses for all sessions were matched to the yellow-tinted dotted line denoting the extent of the place cells recorded during the robot session. (**B**) Left panel: the mean success rate of pellet acquirement (± SEM) during pre-robot, robot, and post-robot session (****p<0.0001, n = 32 recording days from four rats). Right panel: the mean outbound foraging time (± SEM) from the gate opening for animals to reach the pellet (pre-robot and post-robot sessions) or the robot trigger location (robot session) (****p<0.0001, n = 32 recording days from four rats). Each circle represents each recording day's data. (**C**) A schema of simultaneous recordings (left) and photomicrographs of tetrode tips in BA (middle) and dHPC (right). (**D**) Simultaneously recorded 1999 BA-dHPC unit pairs were evaluated at the four different time epochs: 2.5 s before each robot activation (pre-surge; robot session), 2.5 s after each robot activation (post-surge; robot session), 2.5 s before each pellet procurement (pre-pellet; pre-robot session), and 2.5 s after each pellet acquirement (post-pellet; pre-robot session). (**E**) Cross-correlograms (CCs) of all BA-dHPC cell pairs (n = 80) that showed significant spike synchrony during the pre-surge epoch (left) but not during the post-surge (middle) and pre-pellet (right) epochs. CCs of a representative pair are shown above each epoch data, 10 ms bin (the gray shaded area indicates the time window for statistical significance, –100 ms to +100 ms from the BA spikes). The vertical lines (0) indicate the time when the reference BA spikes occurred. The horizontal lines indicate the borders between the presumable dHPC→BA pairs (above the line) and BA→dHPC pairs (below the line). (**F**) The mean firing rates of BA and dHPC cells showed significant spike synchrony during the pre-surge epoch (the gray shaded area, BA = 42 cells, dHPC = 63 cells; overlapping cells were counted once) and the mean speed of the animals (n = 20 recording days from three rats). The dark lines and shaded bands represent the mean and SEM. (**G**) CCs of all BA-dHPC cell pairs (n = 130) that showed significant spike synchrony during the post-surge epoch (middle) but not during the pre-surge (left) and post-pellet (right) epochs. CCs of a representative pair are shown above each epoch data. (**H**) The mean firing rates of BA and dHPC cells showed significant spike synchrony during the post-surge epoch (the gray

*Figure 1 continued on next page*

*Figure 1 continued*

shaded area, BA = 64 cells, dHPC = 102 cells; overlapping cells were counted once) and the mean speed of the animals (n = 20 recording days from three rats). LA: lateral amygdala; CE: central amygdala.

The online version of this article includes the following figure supplement(s) for figure 1:

**Figure supplement 1.** Simultaneous recordings from the basal amygdala (BA) and dorsal hippocampus (dHPC).

applied to exclude the chance of false correlations due to covariation or nonstationary firing rate from the BA and dHPC (see Materials and methods for the detailed steps for the CCs analyses). All corrected CCs were then identified by the following criteria (*Kim et al., 2018*): (i) the average firing rate during each epoch was >0.1 Hz in both BA and dHPC cells; (ii) CCs showed significant peaks, which exceeded the Z-score of 3; and (iii) the peak Z-score fell into a −100 ms and +100 ms time window around the reference BA spikes. Given both direct and indirect projections between the two regions (*Petrovich et al., 2001*; *Pitkanen et al., 2000*; *Rei et al., 2015*; *Saunders et al., 1988*; *Wang and Barbas, 2018*), the ±100 ms time window was used for the spike synchrony and projection directionality (*Burgos-Robles et al., 2017*; *Kim et al., 2018*; *Narayanan and Laubach, 2009*; *Wirtshafter and Wilson, 2020*).

Among all simultaneously recorded 1999 pairs, 714 pairs met the minimum firing rate requirement. From 714 pairs, 30% of pairs (n = 210) showed significant synchrony during pre- or post-surge epochs. During the pre-surge epoch, 80 pairs showed significant spike synchrony (*Figure 1E*, left panel, 11.2%). The same pairs, however, did not show correlated firing during post-surge and pre-pellet epochs (*Figure 1E*, middle and right panels). Another subset of BA-dHPC cell pairs (n = 130, 18%) showed significant spike synchrony during the post-surge epoch, but not during the pre-surge and post-pellet epochs (*Figure 1G*). There were only seven pairs (3%) that showed significant synchrony during both pre- and post-surge epochs (*Figure 1—figure supplement 1C*), and the different sets of neuronal pairs showed distinct firing patterns during the robot interactions (*Figure 1F and H* and *Figure 1—figure supplement 1C*). While there were also BA-dHPC cell pairs that showed distinct synchrony during the pellet procurement (n = 175 pairs; *Figure 1—figure supplement 1D and E*), only 29 pairs revealed synchrony to both aversive robot and appetitive pellet experiences (*Figure 1—figure supplement 1F*). Altogether, these data indicate that distinct subsets of BA-dHPC cell pairs communicate strongly and specifically either before the robot surge (presumably when the animal was facing the robot) or after the robot attack during the risky robot session, but not during the safe pellet procurement trials.

We then explored the directionality of spike synchrony during the robot encounters (*Figure 1E and G* and *Figure 1—figure supplement 1G–I*). If the dHPC spike peak was between −100 ms and 0 ms from the BA spikes, this pair was identified as a dHPC leading pair (dHPC→BA), whereas if the dHPC spike peak was between 0 ms and +100 ms from the BA spikes, the pair was defined as a BA leading pair (BA→dHPC). From the pairs that showed significant spike synchrony during the pre-surge, 40 pairs were identified as dHPC→BA (above the horizontal line in *Figure 1E*, pre-surge), and 40 pairs were identified as BA→dHPC (below the horizontal line in *Figure 1E*, pre-surge). The correlated spikes were significantly higher during the pre-surge than post-surge and pre-pellet in both directions (*Figure 1—figure supplement 1G*). Among the pairs that showed significant spike synchrony during the post-surge, 59 and 71 pairs were identified as dHPC→BA (above the horizontal line in *Figure 1G*, post-surge) and BA→dHPC (below the horizontal line in *Figure 1G*, post-surge), respectively, and their correlated firing was significantly higher during the post-surge than the pre-surge and post-pellet (*Figure 1—figure supplement 1H*). The proportions of the dHPC→BA and BA→dHPC cell pairs were not different between the pre-surge and post-surge peaked CCs (*Figure 1—figure supplement 1I*) with the increased number of significant pairs in both directions during the post-surge epochs (from 80 to 130). These data suggest that a subset of BA and dHPC cells show selective synchronizations during the predatory interaction.

## BA and dHPC cell heterogeneity and their dynamic interaction during risky foraging

To determine whether heterogeneous encoding of the predatory threat situation in the BA and dHPC neurons could differently shape the spike synchrony between the two regions, we categorized

simultaneously recorded 250 BA putative pyramidal cells (*Figure 2—figure supplement 1B*) and 319 dHPC place cells based on their responses to the robot (BA cells) and firing locations (dHPC cells).

Amongst 250 BA cells, 19% and 15% of cells exhibited differential firing characteristics (either excited or inhibited, *Supplementary file 1A*) exclusively to the robot (*Robot cells*, n = 47; *Robot-excited cells,* 45 cells, *Figure 2A and B*; *Robot-inhibited cells*, 2 cells, *Figure 2—figure supplement 1D*) and pellet (denoted as *Pellet cells*, n = 37, *Figure 2—figure supplement 1D*), respectively. Another subset of BA cells (10%) responded to both the robot and pellet (denoted as *Robot + Pellet* cells, n = 25, *Figure 2—figure supplement 1D*), and the rest of the BA cells were not responsive to either the robot or pellet (denoted as *non-responsive cells*, n = 141, 56.4%, *Figure 2—figure supplement 1D*). The non-responsive cells and the Pellet cells were categorized as '*nonRobot*' cells. The Robot + Pellet cells were not included in the further analyses to exclusively compare robot-responsive and robot-non-responsive cells during the robot surge. Interestingly, BA Robot cells may also continue to convey threat information to output structures, perhaps to prepare various anti-predatory defensive behaviors, as they exhibited sustained activities that persisted ~10 s after the robot activations (*Figure 2—figure supplement 1C*), which is much longer than ~2 s duration reported in lateral amygdala (LA) neurons (*Kim et al., 2018*).

Based on our previous report that the stability of hippocampal place cells decreases as a function of distance from the safe nest, 319 dHPC place cells were classified into three cell types by the location of maximal firing during the pre-robot session (*Figure 2C and D* and *Supplementary file 1B*): inside the nest (*nest cells*, n = 213), near the nest (*proximal cells*, n = 25), and afar the nest (*distal cells*, n = 81). Consistent with previous findings, the distal place cells showed less stable firing during the robot session than the nest and proximal cells, as evidenced by the lower spatial correlation (Z') and the higher peak distance between the pre-robot and robot sessions. In the same way, the spatial correlations and peak distances across the sessions were negatively and positively related to the X positions of the place fields, respectively (*Figure 2E*). Furthermore, when animals faced the robot-predator (from –2.5 s to 2.5 s of robot activation), distal cells showed increased theta frequency (6–10 Hz) power compared to nest cells (*Figure 2—figure supplement 2D*). Selective remapping of distal cells during the robot-predator interaction is unlikely due to simple sensory or motor processing per se given the (i) absence of novelty- or sensory-related responses (*Appendix 1—figure 1A-C* ), (ii) the transient residual effects of the robot experience on the stability of place cells during the earlier trials of the post robot session (*Appendix 1—figure 1D and E*), and (iii) no correlation between the speed changes and spatial correlation (*Appendix 1—figure 1F*). For detailed place cell analysis, see Appendix 1.

Next, we explored the synchrony dynamics between the different cell types within the BA and dHPC cells and the effects of the cell-type-specific synchrony on spatial representations of the risky foraging situation in dHPC place cells. To do so, the cell pairs that showed significant spike synchrony during the pre-surge epoch or post-surge epoch were further categorized into different BA (Robot and nonRobot)-dHPC (nest + proximal and distal) cell-type pairs (*Figure 2—figure supplement 3A*). Nest and proximal cells were combined in succeeding spike synchrony analyses because there was no difference in spatial correlations between these two cell types (*Figure 2D*).

From sub-categorized cell pairs, we investigated whether the stability of dHPC cells was dissimilarly altered when they were paired with different types of BA cells (Robot vs. nonRobot cells). When spatial correlations between the pre-robot and robot sessions were examined, the distal cells that showed significant spike synchrony with BA Robot cells during either the pre- or post-surge epoch were found to remap greater than other distal cells correlated with BA nonRobot cells (*Figure 2F–H*). These selective effects on distal cells by the BA cell types were not observed in nest + proximal cells. Also, when the spatial correlations between the pre-robot and post-robot sessions were analyzed, this selective reduction of the spatial correlation was found in the distal cells synchronized with Robot cells during the post-surge (not pre-surge) epoch (*Figure 2—figure supplement 3C*), indicating possible residual effects consequent to the encounter with the predatory robot. When the spatial correlations of the place cells firing together with Robot or nonRobot BA cells were plotted by X positions of the place fields, only the Robot cell-paired, but not nonRobot cell-paired, place cells exhibited decreases in Z' values as a function of the distance from the safe nest (*Figure 2F*). Regardless of the directionality, distal cells both leading and led by Robot cells remapped more during the robot session compared with those paired with nonRobot cells (*Figure 2—figure supplement 3D*). In addition,

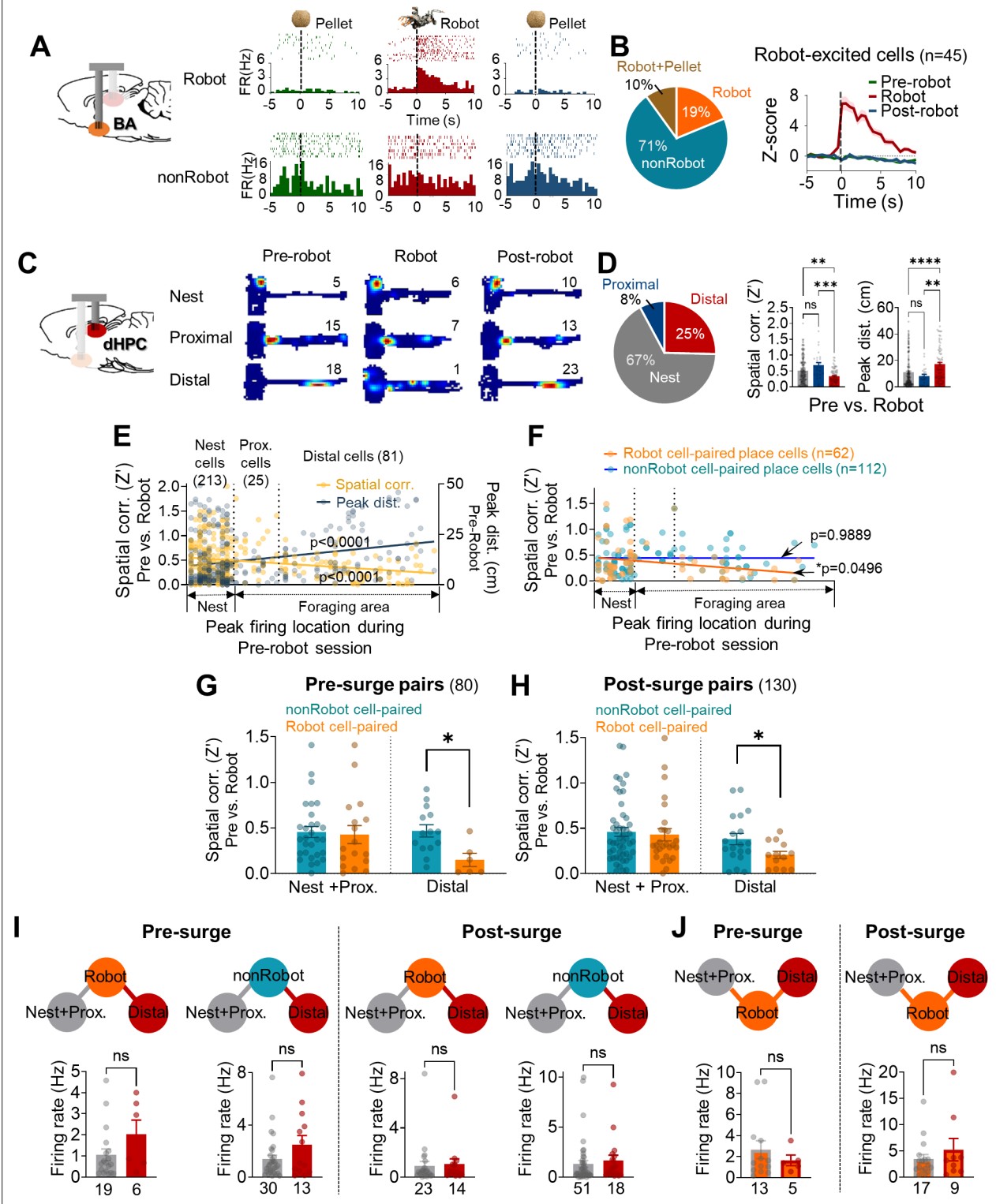

**Figure 2.** Spike synchrony between dorsal hippocampus (dHPC) units and basal amygdala (BA) units during the robot-predator encounter. (**A**) The BA raster plots and peri-event histograms (PETHs) of representative Robot cell (top) and nonRobot cell (bottom) during pre-robot, robot, and post-robot sessions. (**B**) The percentage of different categories of BA cells (left) and the normalized population activity of all robot-excited cells during all three sessions (n = 45). (**C**) dHPC place fields from the nest, proximal, and distal cells during pre-robot, robot, and post-robot sessions (the numerical value represents the peak firing rate). (**D**) Left: the percentage of three place cell types. Middle: the pixel-by-pixel spatial correlation (**Z'**) values between the pre-robot and robot sessions of three place cell types (**p=0.0062, ***p=0.0003, nest = 213 cells, proximal = 25 cells, distal = 81 cells). Right: the

*Figure 2 continued on next page*

*Figure 2 continued*

peak distances between the pre-robot and robot sessions of three place cell types (**p=0.0047, ****p<0.0001). (**E**) Spatial correlations (yellow, *r* = −0.2203, p<0.0001) and peak distances (navy, *r* = 0.2594, p<0.0001) of all place cells between pre-robot and robot sessions are plotted as a function of the peak firing location during the pre-robot session (left, nest; right, end of the foraging distance; circles individual data with regression lines). (**F**) Spatial correlations (pre-robot vs. robot sessions) of place cells that co-fired with Robot (orange circles) or with nonRobot (blue circles) cells are plotted as a function of the peak firing location during the pre-robot session (62 Robot cell-paired place cells, linear regression, *r* = −0.2446, *p=0.0496; 112 nonRobot cell-paired place cells, linear regression, *r* = −0.001316, p=0.9889). (**G**) The spatial correlations between the pre-robot and robot sessions of the nest + proximal cells paired with Robot vs. nonRobot cells during the pre-surge (p=0.7961) and the distal cells paired with Robot vs. nonRobot cells during the pre-surge (*p=0.0119). (**H**) The spatial correlations between the pre-robot and robot sessions of the nes + proximal cells paired with Robot vs. nonRobot cells during the post-surge (p=0.5939) and the distal cells paired with Robot vs. nonRobot cells during the post-surge (*p=0.0430). (**I**) The mean firing rates between the Robot cell-paired nest + proximal cells and the Robot cell-paired distal cells during the pre-surge epoch (the first and second graphs) and during the post-surge epoch (the third and fourth graphs). The numeric values represent the number of cells in each cell type. (**J**) The mean firing rates of nest + proximal or distal cell-paired Robot cells during the pre-surge epoch (left) and during the post-surge epoch (right). The numbers below each graph represent the number of cells, and each circle represents individual cell data. Data are presented as mean ± SEM.

The online version of this article includes the following figure supplement(s) for figure 2:

**Figure supplement 1.** The basal amygdala and the risky foraging behaviors.

**Figure supplement 2.** The dorsal hippocampus and the risky foraging behaviors.

**Figure supplement 3.** Simultaneous recordings from the basal amygdala (BA) and dorsal hippocampus (dHPC) during the risky foraging behaviors.

**Figure supplement 4.** The relationships between firing rates and cross-correlations.

more Robot→distal cell pairs displayed significant synchronous firing during the post-surge than pre-surge epoch, whereas the proportions of the distal→Robot cell pairs did not differ across the pre- and post-surge epochs (*Figure 2—figure supplement 3E*); this indicates that predator attacks caused more robot-responsive BA cells to convey signals to the distal hippocampal place cells. Altogether, these data suggest that BA cells encoding imminent threat might be engaged in distance-dependent spatial representations by closely firing with dHPC distal place cells and destabilizing their activities.

It is possible that the effect of spike synchrony on the spatial correlations simply derived from enhanced firing rates in Robot cells and distal cells during the robot session, which increased the 'chance' of detecting spike synchrony, rather than due to functional interactions between fear encoding BA cells and place encoding dHPC cells. To examine the likelihood of 'co-firing modulation effects on cross-correlations and spatial correlations,' we compared the firing rates of cells that showed significant spike synchrony during pre- and post-surge epochs. First, additional analyses revealed that BA and dHPC cells did not display time-locked responses to robot activation, and the peak response latencies following the robot surge did not overlap between the BA and dHPC cells, especially between Robot cells and distal cells (*Figure 1F and H*, *Figure 2—figure supplement 4A and B*). Second, while BA Robot cells increased firing compared to nonRobot cells (*Figure 2—figure supplement 1D*, *Figure 2—figure supplement 4C*), there were no reliable differences between Robot cell-paired distal cells vs. nest + proximal cells in their firing rates either during the pre-surge or post-surge period (*Figure 2I*). Also, although there was no difference in firing rates between Robot cells paired with nest + proximal and distal cells (*Figure 2J*), only distal cells showed significantly reduced spatial correlations between the pre-robot and robot sessions (*Figure 2—figure supplement 4D*). Lastly, 'Robot + Pellet BA-dHPC' cell pairs that showed significant synchrony during the robot session (pre- and post-surge epochs) did not show reliable synchrony during the pre-robot session (pre- and post-pellet epochs, *Figure 2—figure supplement 4E and F*), even though Robot + Pellet cells had compatible firing rates during the pre-robot and robot sessions (*Figure 2—figure supplement 4E and G*). Collectively, these results suggest that the increase in spike synchrony in BA-dHPC cells cannot be ascribed to increased firing rates.

## Optogenetic stimulation of the BA and defensive behaviors

Based on the findings that the BA cell signaling of a predatory robot strongly associated with the dHPC place cell firing less stably near the threat location (present study) and that neither the foraging (e.g., appetitive, motor) behavior nor the distal place field stability was disrupted by the surging robot in amygdala lesioned/inactivated rats (*Choi and Kim, 2010*; *Kim et al., 2015*), we investigated the causal role of BA activation on the stability of dHPC place cells. To do so, we first confirmed that optogenetic stimulation effectively altered neural activities in the virus-infected

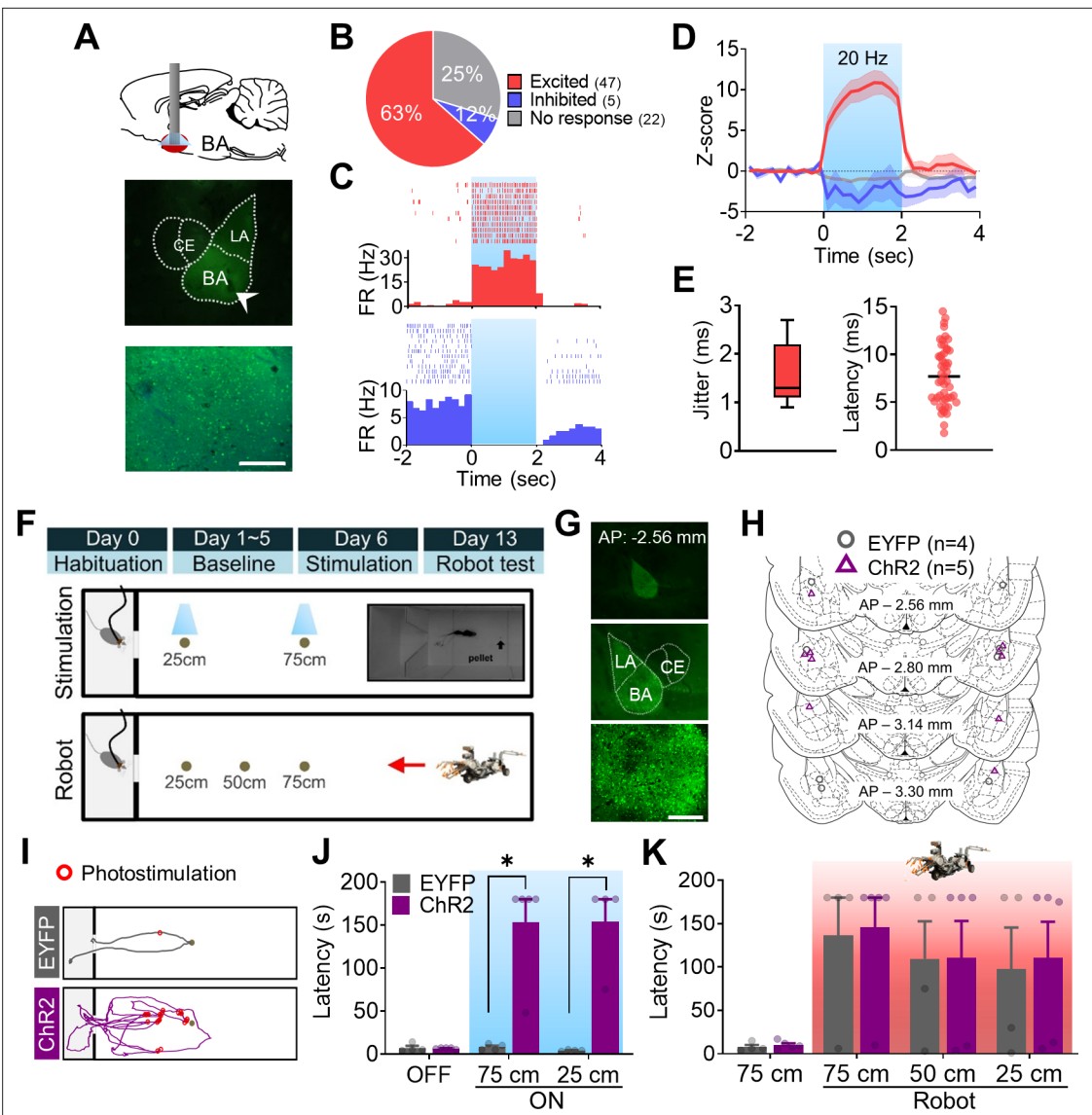

**Figure 3.** Optogenetic manipulations of the amygdala alter the foraging behaviors of naïve rats in the absence of external threats. (**A**) A schema of optic fiber implant (top) and a photomicrograph of optic fiber tip in the basal amygdala (BA; middle) and overlaid image of EYFP and DAPI (bottom). (**B**) The percentage of cells that responded differentially to the photostimulation. (**C**) Peri-event histogram (PETH) and raster plot of a representative excited cell (upper) and inhibited cell (bottom) in response to the photostimulation. The blue shaded area indicates the photostimulation period (2 s, 20 Hz). (**D**) Z-scored firing rates of each cell type (red, excited; blue, inhibited; gray, no response). The dark lines and shaded bands represent the mean and SEM, respectively. (**E**) Jitter and latency of light-evoked responses. (**F**) Illustrations of the experimental design. A pellet was set at 25 cm, 50 cm, or 75 cm distance per trial (inset: the actual foraging apparatus). (**G**) A representative viral expression in the BA at different magnifications. (**H**) Placements of optic fiber tips bordering above or within the BA. Gray and purple circles indicate EYFP-expressing (n = 4) and channelrhodopsin (ChR2)-expressing (n = 5) rats, respectively. (**I**) Representative trajectory plots of EYFP- and ChR2-expressing rats. Red circles indicate the stimulation delivery locations during a 75 cm distance trial. (**J**) The latency of procuring pellets without photostimulations (OFF, p=0.7843) and with photostimulations during the 75 cm (ON-75 cm, *p=0.0159) and 25 cm (ON-25 cm, *p=0.0286) distance trials. (**K**) Latency of procuring pellets during the Robot test. The red shaded area indicates the trials with the robot-predator. Data are presented as mean ± SEM, and individual data are represented as distinct symbols. LA: lateral amygdala; CE: central amygdala. Scale bars, 200 μm.

The online version of this article includes the following figure supplement(s) for figure 3:

**Figure supplement 1.** Locomotor data during the optogenetic stimulation.

cells (*Figure 3A*). Specifically, in four animals injected with channelrhodopsin (ChR2)-expressing adeno-associated viruses (AAVs) and implanted with an optrode in the BA (*Figure 3—figure supplement 1A*), we found that light stimulation increased spiking activity in 47 cells (Z > 3), decreased spiking activity in 5 cells (Z < –3), and produced null effects on spiking activity in 22 cells in the BA (*Figure 3B–D*). The enhanced firing to light stimulation was reliable throughout the recording session, confirming the efficacy of optogenetic manipulation (*Figure 3E*). We then tested whether the optogenetic stimulation of the BA induces defensive behaviors in naïve rats sans a predatory robot. ChR2 (n = 5) or EYFP (AAV-EYFP; n = 4) expressing rats received 2 s photostimulation each time they approached the pellet (approximating the robot trigger distance in *Figure 1* experiment, ~ 25 cm from the pellet). After the optogenetic test, all rats were challenged with the robot-predator without the photostimulation (*Figure 3K*). As can be seen, the virus expression was limited to the BLA (*Figure 3G*), and optic fiber tips were in the middle part of the BLA (*Figure 3H*). Without the photostimulation, ChR2 and EYFP rats were able to procure the pellet successfully (*Figure 3J*, OFF). With the stimulations, however, ChR2 animals exhibited significantly longer procurement latencies than EYFP animals to both pellets at 75 cm (*Figure 3J*, ON-75 cm) and 25 cm (*Figure 3J*, ON-25 cm). Because each photostimulation ON and succeeding OFF events resulted in fleeing and foraging behaviors, respectively, ChR2 rats traveled more distance and received a greater number of photostimulation compared to EYFP rats (*Figure 3—figure supplement 1B, C, F and G*). The BA photostimulation also induced pause and retract defensive behaviors in the absence of external threat (*Figure 3—figure supplement 1D and H*). These effects were not due to tissue damage in the BA incurred by repetitive light stimulations because both ChR2 and EYFP groups fled from the looming robot-predator and consequently took a prolonged time to acquire the pellet (*Figure 3K*). These results indicate that increased BA pyramidal neuronal activities can elicit robust defensive (fear) behaviors in foraging animals even when there is no explicit threat in the environment.

## Optogenetic stimulation of the BA and dHPC place cell activities

We next examined whether optogenetic stimulation of the amygdala can sufficiently alter the stability of hippocampal place cells. A separate group of rats (n = 8), with ChR2 in BA and tetrodes in dHPC (ipsilateral), underwent three successive sessions of pre-stimulation, stimulation, and post-stimulation (8–10 trials/session), which were analogous to pre-robot, robot, and post-robot sessions. A small number of cases where the stimulation failed to elicit fleeing behavior due to inaccurate optic fiber placement (from one animal; one recording session) or low light intensity (<5 mW, from one animal; five recording sessions) provided an opportunity to evaluate activities of place cells with (*Figure 4A–E*, *Figure 4—figure supplement 1A–H*, and *Supplementary file 1C*) and without (*Figure 4F–J*, *Figure 4—figure supplement 1I–L*, and *Supplementary file 1D*) behavioral effects. Specifically, rats with behavioral effects made multiple attempts to get a pellet during the stimulation session (*Figure 4A*, middle), yet the success rate was vastly lower compared to pre- and post-stimulation sessions (*Figure 4A*, right). Those that did not respond to the stimulation promptly procured the pellet during the stimulation session (*Figure 4F*). The dHPC place cells from the rats with or without behavioral effects were classified into three types (with behavioral effects: *nest* = 125, *proximal* = 21, *distal* = 107; without behavioral effects: *nest* = 30, *proximal* = 8, *distal* = 26; *Figure 4B and G*) in the manner described previously. Spatial correlations and peak distances between pre-stimulation and stimulation sessions decreased and increased, respectively, in distal cells compared to nest cells when the photostimulation caused rats to escape into the nest (*Figure 4C*, *Figure 4—figure supplement 1D and E*). These effects on distal cells were absent in rats without stimulation-induced behavioral effects; that is, both spatial correlations and peak distances of the distal cells were not different from those of the nest and proximal cells (*Figure 4H*, *Figure 4—figure supplement 1K and L*). Furthermore, spatial correlations were negatively correlated with the peak firing locations from the nest, while peak distances were positively correlated with the peak firing locations. The inverse relationships between the X positions vs. spatial correlations and the X positions vs. peak distances were found only in rats that showed defensive behaviors in response to the photostimulation (*Figure 4D and I*). Consistent with our present and previous findings that only distal cells showed increased theta frequency (6–10 Hz) power during the robot session (*Kim et al., 2015*), optogenetic stimulation of the BA also increased theta power selectively in distal cells of fleeing rats during the 5 s epochs subsequent to photostimulation (*Figure 4E*).

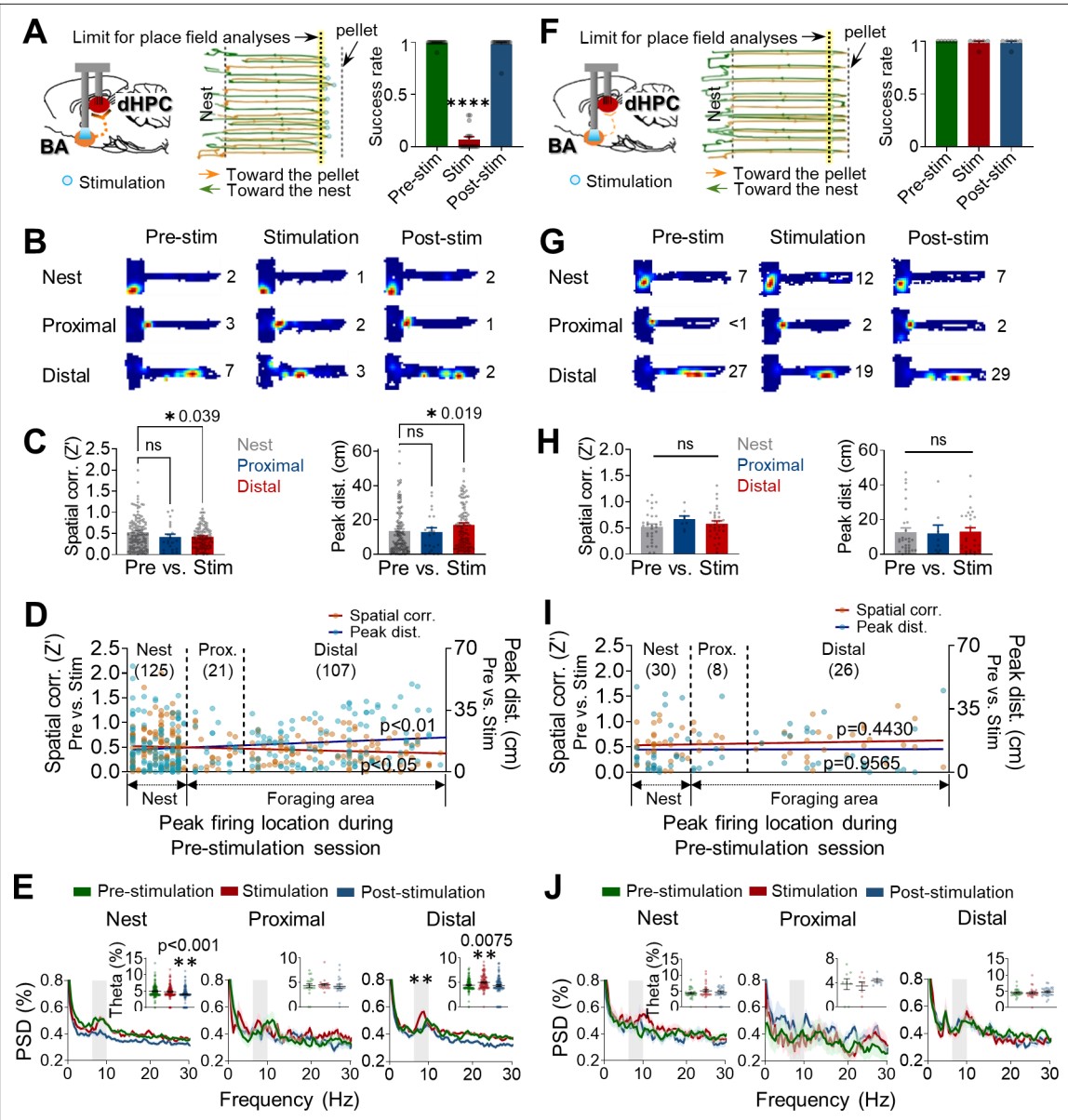

**Figure 4.** Optogenetic stimulations of the amygdala alter the stability of place cells. (**A–E**) Data were collected from rats exhibiting defensive behaviors responding to the photostimulation. (**A**) An illustration showing that optogenetic stimulation of the basal amygdala (BA) presumably sends strong (primarily polysynaptic) inputs to the dorsal hippocampus (dHPC) when the stimulations elicited behavioral effects (left). A representative tracking plot during the stimulation session (middle). Orange lines and blue circles indicate the outbound trails and stimulation locations, respectively, that evoked escape responses (green lines). The yellow-tinted dotted line represents the limit of place field analyses. The success rate of pellet retrieval during the photostimulation session significantly decreased (right, n = 17 recording days, ****p<0.0001). (**B**) Examples of place fields from the nest, proximal, and distal cells during each session (the numerical value represents the peak firing rate). (**C**) Differences in the pixel-by-pixel spatial correlations (**Z′**) value (left, *p=0.039) and peak distances between the pre-stimulation and stimulation sessions (right, *p=0.019). (**D**) Spatial correlations (individual data: orange squares, regression line: dark red, r = −0.1320, *p=0.0359) and peak distances (individual data: blue circles, regression line: dark blue, r = 0.1625, **p=0.0096) of all place cells between pre-stimulation and stimulation sessions are plotted in the order of X position of rats. (**E**) Power spectral densities (PSD, shown as % of total PSD) of different frequency bands from each cell type during the three sessions. The dark lines and shaded bands represent the mean and SEM, respectively. The gray band represents the theta range (6–10 Hz), and the percentage of theta power during the three sessions is shown in the inserted bar graphs. (**F–J**) The same analyses as in (**A–E**), and data were collected from rats not exhibiting defensive behaviors to the photostimulations. (**F**) When the stimulation of the BA was too weak (left) to elicit defensive behaviors (middle), the success rates across the three sessions were not different (right, n = 6 recording days, p>0.999). (**I**) Spatial correlations: r = 0.0.09758, p=0.4430; peak distances r = 0.0.006948, p=0.9565. Data are presented as mean ± SEM.

The online version of this article includes the following figure supplement(s) for figure 4:

**Figure supplement 1.** Basal amygdala (BA) photostimulation effects on place cell stability.

In rats without photostimulation-induced behavioral effects, theta power did not increase in distal cells (*Figure 4J*).

Because selective remapping of the distal cells could be a result of direct modulations in firing rates by the optical stimulation rather than by the stimulation-elicited emotional state. We further analyzed how dHPC cells responded to the optical stimulations by measuring stimulation-evoked firing rate changes (Z-scores) during the 1 s stimulation period. Although a subset of cells (in all three cell types) were directly modulated by the stimulations ($Z > 3$ for excited and $Z < –3$ for inhibited responses; *Figure 4—figure supplement 1F*), there was no significant difference in either the firing rate changes during the 1 s stimulation across the three cell types (*Figure 4—figure supplement 1G*) or the spatial correlations between stimulation-neutral and stimulation-responsive cells (*Figure 4—figure supplement 1H*). These results suggest that although BA stimulation elicited firing changes in some of the dHPC cells, the firing alteration did not directly cause changes in the spatial tuning of place cells. Taken together, these results are consistent with the notion that endogenous activation of BA pyramidal neurons disrupted spatial stability of dHPC place cells and impeded successful foraging, mimicking the exogenous predatory threat-induced distal cell remapping and spatial avoidance behavior.

## Discussion

This study recorded, for the first time, simultaneous spike trains from BA and dHPC cells while rats were foraging for food, a purposive behavior (*Tolman, 1948*), in a risky predatory situation that virtually all animals are likely to encounter in the wild (including prehistoric humans; *Mithen, 1999*), which differs significantly from the standard Pavlovian fear conditioning paradigms measuring a specific response (typically freezing) in a small chamber. As a result, we revealed a novel amygdalar-hippocampal circuit coding mechanism for interfacing danger and place information. Specifically, dHPC place cell activities that synchronized with looming robot-responsive BA cell activities (i.e., increased spiking) exhibited less stable place field properties than those dHPC place cell activities synchronized with robot-unresponsive BA cell activities. More importantly, the robot-responsive BA-dHPC synchrony effects on place fields manifested in areas proximal to the danger and not inside or near the safe nest, possibly representing brain activity subserving a spatial gradient of fear. Furthermore, there was a concomitant increase in theta rhythm power in the distal areas where the predatory robot was positioned and the place field stability was most altered, a finding consistent with a report of high levels of synchronized theta in the amygdalo-hippocampal network in fear-conditioned animals (*Narayanan et al., 2007*; *Seidenbecher et al., 2003*). The possibility that amygdala-coded fear can uncouple theta rhythm and place field stability in the hippocampus is supported by the findings that optogenetic BA stimulation was sufficient to elicit spatial avoidance behavior, increase the theta rhythm power, and disrupt the stability of place fields in the absence of external agent of danger. It follows then the crucial factor that influences the place field stability is the locus of amygdala-coded fear and not the foraging distance itself, and this can be tested in future studies by eliciting fear in proximal, but not distal, foraging distances (e.g., using a two pellet location choice foraging task; *Kim et al., 2016*). While the majority of amygdala and dHPC projections is polysynaptic, via dorsal CA3 and ventral HPC areas (*McDonald and Mott, 2017*; *Pikkarainen et al., 1999*; *Rei et al., 2015*), there is also evidence of sparse direct amygdala and dHPC projections (*Petrovich et al., 2001*; *Pitkanen et al., 2000*; *Wang and Barbas, 2018*). The relative contributions of polysynaptic vs. monosynaptic amygdala-dHPC projections to the present results, however, require further research employing circuit-specific and genetically defined cell-type-specific manipulations in transgenic mice models (e.g., selective stimulations of retrogradely labeled BA neurons that sparsely project to the dHPC place cells or multi-step stimulations/recordings including the di-synaptic BA-CA3/vHPC-dHPC circuits). Moreover, while the optogenetic stimulation results may be consistent with the notion that endogenous activation of BA pyramidal neurons disrupted spatial stability of dHPC place cells and impeded successful foraging, a major caveat of our single-site stimulation approach is that neither the possibility of non-specific stimulation effects nor involvement of other brain regions can be excluded. The latter possibility, however, is unlikely given that amygdalar lesions effectively blocked predatory robot-induced fear and remapping of dHPC place cells (*Kim et al., 2015*).

There are three main confounding variables and alternative explanations in the study that need to be considered. First, the spike synchrony in BA and dHPC (i.e., distal place) cells in a risky foraging situation could simply be a by-product of increased firing rates, rather than functional interactions,

in BA and dHPC cells to the robot-predator. The firing modulation effect, however, is unlikely based on the shift-predictor correction analysis (i.e., shuffling trials of the reference and target neurons), the unmatched BA and dHPC responses to the predatory event (*Figure 1F and H*), the similarly increased dHPC firing rates in nest, proximal and distal foraging regions (with remapped place fields limited to distal place cells), and the comparable firing rates between Robot cell-paired distal cells (with remapped place fields) vs. nonRobot cell-paired distal cells (with stable place fields) (*Figure 2G and H*). Second, the present findings may be due to novelty/saliency features of the robot (rather than robot-evoked fear) signaled by the amygdala. This possibility is also unlikely given that distal place fields were stable with a stationary robot that does not elicit fear (*Appendix 1—figure 1A*), and that amygdala-lesioned rats unafraid of (i.e., did not flee from) the surging robot exhibited stable distal place fields and normal theta rhythm power and appetitive (hunger motivated) behavior (*Kim et al., 2015*). The latter study also found that in amygdala-intact rats the distal cell remapping was observed in animals that failed to procure pellets (high-fear state) but not in those that sporadically showed successful foraging (low-fear state) to the looming robot. Furthermore, there is evidence of amygdala-lesioned rats exhibiting normal novel object recognition memory (*Aggleton et al., 1989*; *Mumby and Pinel, 1994*) and displaying normal learning and enhanced memory of a visually salient platform water maze task (*Kim et al., 2001*). Lastly, it is generally held that the firing rate of place cells correlates positively with the running speed of animals pursuing food freely sans danger in the recording environment (*Czurko et al., 1999*; *Lu and Bilkey, 2010*; *McNaughton et al., 1983*). Thus, the place cell remapping observed in this ecologically relevant 'approach food-avoid predator' study may reflect velocity differences in proximal vs. distal regions rather than fear magnitude differences. However, a previous study examined the relationship between place cell firing property and the running speed directionality of movement (i.e., proximal-outward speed, distal-outward speed, distal-inward speed, proximal-inward speed) during the robot session and found that the differential stability of place fields cannot be accounted solely by the speed change or acceleration (*Kim et al., 2015*), and the current study also confirmed that speed changes by the surging robot did not selectively affect stabilities of distal cells the same (*Appendix 1—figure 1E*). Specifically, there was no significant correlation between the relative speed and spatial correlation in both nest + proximal and distal cells, suggesting that hippocampal remapping during the robot session cannot be fully explained by the running speed per se. Furthermore, the fact that place cell remapping was observed across fear conditioning (*Moita et al., 2003*; *Moita et al., 2004*), inhibitory avoidance (*Schuette et al., 2020*), and ethological fear (*Kim et al., 2015*) paradigms, where the animals exhibited dissimilar fear behaviors, strongly suggests that fear at least partly contributes to alterations in place fields.

Amir, Pare, and colleagues used a similar 'approach food-avoid predator' paradigm (*Choi and Kim, 2010*) and found that neuronal activities in the basolateral amygdala (BL), a region corresponding to BA, correlated with the animals' movement velocity whether they were foraging in 'no-robot' days or foraging in 'robot' days (*Amir et al., 2019*; *Amir et al., 2015*). Specifically, most BL neurons were found to be inhibited during foraging and to the predator, and only 4.5% cells showed predator-responsive activities. Thus, they concluded that the amygdala activity aligned largely with 'behavioral output' (i.e., foraging) rather than with threats/fear. In the present study, however, the majority of BA neurons showed dissimilar responses during the pre-robot and robot sessions. Predatory threat/fear-responsive neuronal activities were also identified in the LA (*Kim et al., 2018*). What, then, can account for the apparent discrepancy in the findings when both groups employed an ecologically relevant paradigm? There were obvious differences in the apparatus features (e.g., size), experimental procedures (e.g., pellet locations), and unit data analysis (see *Amir et al., 2019* for details), which could have led to different results and conclusions concerning the significance of amygdalar neuronal activities. It is also worth noting that the present study tracked the same amygdalar neurons during pre-robot and robot sessions, allowing direct comparisons of neuronal activities in the presence and absence of predatory threat. Notably, there was a significant difference in the foraging success rate, that is, ~80% (*Amir et al., 2015*) vs. <3–4% (the present study and *Kim et al., 2018*). If the pellet procurement rate inversely correlates with the fear magnitude, then the high foraging success rate associated with inhibited amygdalar activity (*Amir et al., 2015*; *Amir et al., 2019*) can be inverted by disinhibiting the amygdalar activity, while the low foraging success rate associated with increased amygdalar activity (present study) can be reversed by suppressing the amygdalar activity. The former prediction is consistent with the present findings that optogenetic stimulation of BA neurons per se

**Figure 5.** A hypothetical coding model of the safe-danger boundary by the amygdala-hippocampus network. (**A**) Illustrations of amygdala and hippocampus cell pairs that showed synchronized firings when the animal is confronted with the robot-predator: Robot-Distal (left) and non-Robot-Distal (right) pairs. (**B**) An illustration of spatial representation of the safe-danger boundary (the gray area) in the hippocampus as an outcome of interaction with the amygdala. The concentric circles, outer amygdala cells (Robot or nonRobot), and inner hippocampal place cells (nest/proximal or distal) represent safe vs. dangerous environments based on the information from/to the amygdala. Specifically, distal cells synced with Robot cells show a greater extent of remapping (represented as sun-shape), which is presumably due to the eminent fear information received by the amygdala.

caused rats to flee to the nest (and also destabilized hippocampal place fields), whereas the latter prediction is consistent with the previous findings that muscimol inactivation of amygdala prevented dorsal periaqueductal gray stimulation-induced fleeing in the same foraging task (*Kim et al., 2013*).

In nature, all animals are challenged with spatially distributed ecological threats, such as predators and aggressive conspecifics, that they must avoid with proficiency. Although earlier studies (*Kim et al., 2015*; *Moita et al., 2003*; *Moita et al., 2004*) revealed that conditioned (learned) and unconditioned (innate) fear can alter hippocampal neuronal activities, how the brain's cells coordinate risky and place information to generate spatial representation of fear remained unknown. The present simultaneous recording study, empirically anchored to real dangers that animals face in nature, suggests that the synchronous firing between place-coding hippocampal cells and fear-coding BA cells allows constructive foraging behavior. Specifically, the BA cells that immediately respond to predator attacks destabilize place cell firing at the moment/location of the threat, forming spatial gradient of fear so that animals can traverse safe-danger boundaries of their environment (*Figure 5*). It follows, then, that asynchronous activities between hippocampal and amygdala cells may lead to lethal foraging decisions in animals and underlie generalized or context-inappropriate fear disorders in humans by obscuring the safe-danger boundary.

## Materials and methods
### Subjects
Male Long–Evans rats (initial weight 325–350 g) were individually housed in a climate-controlled vivarium (accredited by the Association for Assessment and Accreditation of Laboratory Animal Care), with a reversed 12 hr light/dark cycle (lights on at 7 PM), and placed on a standard food-deprivation schedule with free access to water to gradually reach ~85% normal body weights. All experiments were performed during the dark phase of the cycle in strict compliance with the University of Washington Institutional Animal Care and Use Committee guidelines.

### Surgery
An overview of surgical procedures and experimental descriptions is shown in *Supplementary file 1E*.

#### Simultaneous recording (Figures 1 and 2 experiment)
Under anesthesia (94 mg/kg ketamine and 6 mg/kg xylazine, intraperitoneally), four rats were mounted in a stereotaxic instrument (Kopf) and implanted with a microdrive of tetrode bundles (formvar-insulated nichrome wires, 14 μm diameter; Kanthal) into the right dHPC and BA. Three tetrodes per region were implanted for the two rats, and four tetrodes per region were implanted for the other two

rats. The microdrive was fixed by dental cement with anchoring screws. Behavioral experiments and recording session started after 1 week of recovery.

## Optrode experiment (Figure 3A–E experiment)

Four rats were anesthetized, and virus described below (ChR2) was injected into the right BA via a microinjection pump (UMP3-1, World Precision Instruments) with a 33-gauge syringe (Hamilton). The total volume was 0.5 μl per site, and the injection speed was 0.05 μl/min. To avoid backflow of the virus, the injection needle was left in place for 10 min. After injection, optrode that consists of six tetrodes and one optic fiber with ferrule (0.22 NA, 200 μm core; ferrule diameter: 2.5 mm; Doric Lenses) was implanted in the right BA. The optrode was secured by C&B Metabond and dental cement with anchoring screws. Behavioral experiments started after 4–5 weeks of recovery and viral expression.

## Optogenetic stimulation of the BA (Figure 3F–K experiment)

Under anesthesia, nine rats were injected with one of the viruses described below (ChR2, n = 5; EYFP, n = 4). After the virus injection, optic fibers attached to ferrules were implanted 0.4 mm dorsal to the injection sites. The optic ferrules were secured by Metabond and dental cement with anchoring screws. Behavioral experiments started after 4–5 weeks of recovery and viral expression.

## Optogenetic stimulation of the BA and place cell recording from dHPC (Figure 4 experiment)

Seven rats were anesthetized and mounted in a stereotaxic. The virus described below (ChR2) was delivered into the right BA (n = 5) or bilateral BA (n = 3). Following injection, optic fibers attached to ferrules were implanted into the right BA or bilateral BA (0.4 mm dorsal to the virus injection site). After the virus injection and the optic fiber implantation, a microdrive of tetrode bundles (n = 5) or VersaDrive-8 (n = 3, Neuralynx) was implanted into the dHPC (the same coordinates as used in the simultaneous recording experiment). The optic fibers and electrodes were fixed by Metabond and dental cement with anchoring screws. Behavioral experiments and recording session started after 4–5 weeks of recovery and viral expression.

### Viruses

AAVs (serotype 5) to express Channelrhodopsin-EYFP (AAV5-CaMKIIa-hChR2(H134R)-EYFP, n = 5) or EYFP only (AAV5-CaMKIIa-EYFP, n = 4) were injected in the BA. CaMKII promoter was used for targeting pyramidal neurons favorably in the BA (*Van den Oever et al., 2013*). Viral titers were 8.5 × $10^{12}$ virus molecules/mL for AAV5-CaMKII-hChR2(H134R)-EYFP and 4.3 × $10^{12}$ virus molecules/mL for AAV5-CaMKII-EYFP. Viruses were stored in a –80 °C freezer until the day of surgery. All viruses were obtained from the University of North Carolina Vector Core.

### Behavioral paradigms

Rats maintained their body weights at ~85% of normal weight throughout the sessions. The experiment was conducted in a specialized foraging apparatus (*Kim et al., 2015*). The composition of sessions for each experiment is represented in *Figures 1A, 3F and 4A*.

### Habituation

All rats were placed in the nest for 30 min/day for two consecutive days with 20 food pellets (0.5 g, F0171, Bio-Serv) to acclimate to the nest area and the experimental room.

### Baseline foraging for Figure 1, Figure 3A-E, and Figure 4

Two minutes after the rat was placed in the nest area, the gateway to the foraging area opened and the rat was allowed to explore and procure a food pellet placed at variable distances (25 cm, 50 cm, 75 cm, 100 cm, and 125 cm from the nest). After the rat took the pellet back into the nest, the gateway closed. When the rat learned to procure a pellet from the longest distance, the pellet distance was fixed, and unit screening started. Rats underwent baseline foraging until unit responses were detected.

## Baseline foraging for Figure 3F–K

After 2 min in the nest, the gateway to the foraging area opened, and the rat was allowed to explore and procure a pellet placed at 25 cm from the nest (*Figure 3F*). After the rat took the pellet back into the nest, the gateway closed (first trial). Consecutive trials commenced in the same way, except the pellet distance from the nest increased to 50 cm for the second trial and 75 cm for the third trial. Rats underwent 4–5 days of baseline foraging for the behavioral experiment.

## Behavioral procedure for the simultaneous recording (Figures 1 and 2)

Units from the BA and the dHPC were recorded throughout the three sessions: pre-robot, robot, and post-robot. During the pre-robot session, 8–10 trials of foraging with a pellet placed at 125 cm from the nest were conducted to collect baseline unit activities. After the pre-robot session, rats underwent the robot session with a robot-predator placed at the end of the foraging area (Mindstorms robotic kit, LEGO Systems) (*Choi and Kim, 2010*). After the gateway opened, each time the rat approached the vicinity of the pellet (~25 cm from the pellet), the robot surged 23 cm toward the pellet, snapped its jaws once, and returned to its original position. Rats were permitted at least 10 attempts to procure the pellet. Since one robot activation was counted as one robot trial, the frequencies of visiting the distal zone between the pre-robot and robot sessions were matched. The robot-evoked responses were examined for 10 s after each attempt. If the rats made additional attempts within 10 s following the previous robot activation, those attempts were excluded from the analysis to prevent overlaps in the robot-evoked responses. Once the rats finished the robot session, another 8–10 trials of foraging without the robot were conducted to collect post-manipulation unit activities (post-robot session).

## Behavioral procedure for the optrode recording (Figure 3A–E)

Rats foraged for a pellet placed at 125 cm from the nest while the BA units were recorded. The laser (473 nm; Opto Engine LLC) was connected to Master-8 (A.M.P.I.) to deliver photostimulations (2 s, 20 Hz, 10 ms width, 5–10 mW) and photostimulation-responsive units were detected. Response latency to the photostimulations was calculated for each light pulse (10 ms, 20 Hz, 2 s). An optrode was lowered after each recording session and multiple days of tests were performed. On some of the test days, photostimulations were delivered while rats were under anesthesia. This prevented seizure-like behaviors due to overstimulation of the BA. There was no robot session for the optrode experiment.

## Behavioral procedure for the optogenetic stimulation of the BA (Figure 3F–K)

Once rats learned baseline foraging, the stimulation session began. During the stimulation session, rats first underwent three trials of baseline foraging with a pellet at 125 cm from the nest. For the photostimulation trial, the bilateral BA were activated by laser (2 s, 20 Hz, 10 ms width, 5–10 mW) whenever the rat approached the vicinity (~25 cm) of the pellet. If the pellet was not procured within 3 min, the gateway closed, and the rat was tested with a pellet at 25 cm from the nest for another 3 min. If the rat succeeded with a pellet at longer distance, then a shorter distance pellet testing did not follow (ChR2 group only). For the EYFP group, rats were tested with both 75 cm and 25 cm pellets subsequently. One week after the stimulation session, all rats were tested with the robot-predator. Firstly, rats were allowed to procure the pellet with a pellet at 75 cm without the robot and then underwent the first robot-predator encounter trial with the same distance pellet. If the rat was unsuccessful for 3 min, the pellet was moved to 50 cm and 25 cm distances on the following trials.

## Behavioral procedure for the optogenetic stimulation of the BA and place cell recording (Figure 4)

Place cell activities from the dHPC were recorded throughout the three sessions; pre-stimulation, stimulation, and post-stimulation. During the pre-stimulation session, 8–10 trials of foraging with a pellet placed at 125 cm from the nest were conducted to record baseline place cell activities. After the pre-stimulation session, the bilateral (n = 5) or unilateral (n = 2) BA were stimulated by the laser (1–2 s, 20 Hz, 10 ms width, 1–10 mW) whenever the rats approached the vicinity of the pellet (~25 cm) during the stimulation session. Rats were allowed to attempt to procure the pellet at least 10 times

while the place cell activities were recorded. If the photostimulations were delivered more than twice within 10 s, only the first stimulation was included for the analysis to ensure place cell responsiveness by the photostimulations. Following the stimulation session, rats again were permitted to procure the pellet for 8–10 trials to record post-manipulation effects (post-stimulation session).

## Behavioral data acquisition and analyses

The ANY-maze video tracking system (Stoelting Co.), with an HD webcam (C920, Logitech) affixed over the apparatus was used to capture video images and automatically track the rat's movement (30 frames/s) from both nest and foraging areas. ANY-maze video tracking system was connected to the recording system (Neuralynx) and provided the rat's tracking information. It also provided locomotor data including distance traveled, speed, and the number of entries to specified zones.

## Unit recording and analyses

The impedance of electrode tips was matched to 100–300 kΩ measured at 1 kHz through gold plating. After the postoperative recovery period, electrodes were gradually advanced (≤160 µm per day) until reached the target regions. Unit isolation and cluster cutting procedures have been described before (*Kim et al., 2007*). Briefly, unit signals were amplified (10,000× ), filtered (600 Hz to 6 kHz), and digitized (32 kHz) by using the Cheetah data acquisition system (Neuralynx). Unit isolation was performed by using an automatic spike-sorting program (SpikeSort 3D; Neuralynx) and additional manual cutting as described in previous studies (*Kim et al., 2018*; *Kim et al., 2015*). Raster plots and peristimulus time histograms were generated by NeuroExplorer (Nex Technologies). For all units, we ruled out any chance of recording the same cells across multiple sessions by comparing the shape of the waveform, autocorrelogram, and interspike interval histogram between recording days.

### Place cell analysis

Criteria used for place cell analysis have been described in the previous study (*Kim et al., 2015*). To mention briefly, units that showed (1) stable, well-discriminated complex spike waveforms, (2) a refractory period of at least 1 ms, (3) peak firing >2 Hz in any sessions, and (4) spatial information >1.0 bits per/s in any sessions were included. Based on the peak positions of place fields during the pre-robot/pre-stimulation session, cells were classified into three different types: nest cells (cells fired maximally in the nest), proximal cells (cells fired in the proximal region; the foraging area between 0 and 25 cm from the nest), and distal cells (distant from the nest, close to the threat). A pixel-by-pixel spatial correlation analysis by a customized R program calculated the similarity of the place maps across three different sessions; pre vs. robot/stimulation sessions, robot/stimulation vs. post sessions, and pre vs. post sessions for each place cell. The resulting correlation value (*r*) was converted to a Fisher Z' score for further parametric comparisons between cell types. The customized R program also calculated the distance of each cell's peak firing locations between the sessions. It first finds each cell's maximal firing location (one value that reflects the x-axis of the location; lowest value – the start of the nest, highest value – the end of foraging area) during each session (pre, robot/stimulation, and post sessions). It then calculates the distance (cm) between pre vs. robot/stimulation sessions, robot/stimulation vs. post sessions, and pre vs. post sessions for each place cell.

### Classification of BA units

BA units were classified as putative pyramidal cells and interneurons based on the average spike width and the firing rate of each cell (hierarchical unsupervised cluster, *Figure 2—figure supplement 1B*; *Kim et al., 2018*). The majority of cells recorded from the BA were putative pyramidal cells (n = 250, 96.9%). Due to a small number of interneurons (n = 8, 3.1%), only pyramidal neurons were included in the further analyses. All units' activities were aligned by the event of pellet acquirements (pre- and post-robot sessions), robot activations (robot session), or photostimulations (stimulation session) by using NeuroExplorer (version 5.118, Nex Technologies). All data were binned in time windows of 500 ms. A neuron was defined as Robot-responsive (Robot cell) if the neuronal changes (Z-score >3) exclusively occurred within 0–1.5 s before the robot activation (robot-approaching changes) or within 0–3 s after the robot activation (robot-triggered changes). All unit activities were normalized to the baseline period (–5 s to –1.5 s in the PETH). Pellet cells were classified in the same manner except the test window was –1.5 s to 2 s around the pellet acquirement event. Note that, for the Robot

cell classification, the possibility of neuronal changes to the robot's jaw snapping that occurred at 1.5 s after the robot activation was additionally examined for the test window (0–3 s). Cells that showed significant neural changes (Z-score >3) both to the robot and pellet were classified as Robot + Pellet cells. If a cell did not meet the above classifications, it was classified as a non-responsive cell.

## Cross-correlation

CCs of the simultaneously recorded BA and dHPC units were generated by NeuroExplorer. The analysis procedure was identical as described in the previous study (*Kim et al., 2018*) except that the current study includes 2.5 s of each time epoch (*Figure 1D*). In this analysis, CCs were generated with BA cells as the reference (10 ms bin) with four different time epochs *pre-pellet*, 2.5 s epoch before pellet procurement during the pre-robot session; *post-pellet*, 2.5 s epoch after the pellet procurement during the pre-robot session; *pre-surge*, 2.5 s epoch before the robot looming during the robot session; and (*post-surge*, 2.5 s epoch after the robot looming during the robot session). The total number of pellet procurements was 309 (average of 9.7 trials per recording day; 32 recording days), and the total number of robot activations was 331 (average of 10.3 trials per recording day). To exclude the chance of false correlations due to covariation or nonstationary firing rates from the BA and dHPC, there was a correction and strict criteria for determining significant cross-correlations (*Burgos-Robles et al., 2017*). Specifically, the raw CCs were corrected by 'Shift-Predictor' where 100 times of trial shuffles were applied. Each shift predictor correlogram was subtracted from its respective raw correlogram, and Z-scores were calculated by the mean and standard deviation of the corrected CC. The neural pair was considered to be significantly correlated if the peak Z-score was >3. Additional criteria were that the BA and dHPC firing rates during the pre- and post-surge periods must be above 0.1 Hz, and the peak of the CC should fall within a testing window of ±100 ms relative to the reference spikes.

## Histology

After the completion of the experiment, electrolytic currents (10 µA, 10 s) were applied to each tetrode tips to confirm the placement of the electrodes. Rats were overdosed with Beuthanasia and perfused intracardially with 0.9% saline and 10% formalin. Extracted brains were stored in 10% formalin at 4 °C overnight, followed by 30% sucrose solution until they sank. Transverse sections (50 µm) were washed with phosphate-buffered saline (PBS) and mounted onto slides with the gelatin solution. Staining with Cresyl violet and Prussian blue confirmed the tip locations. To verify viral expression, rats were overdosed with Beuthanasia and perfused intracardially with 250–300 mL of PBS followed by 400 mL 4% paraformaldehyde in PBS. Brains were extracted, stored in 4% paraformaldehyde solution at 4 °C overnight, then transferred to 30% sucrose solution until they sank. Transverse sections (50 µm) were washed with PBS, mounted on to slides, and coverslipped with Flouromount-G with DAPI (eBioscience). The expression of viruses and the location of electrode tips were examined using a fluorescence microscope (Keyence BZ-X800E). Rats with no EYFP expression or misplacement were excluded from the analysis.

## Statistical analyses

Statistical significance was determined with one-way repeated measures ANOVA, two-way ANOVA, linear regression, unpaired *t*-test, Kruskal–Wallis test, or Friedman test using Bonferroni *post hoc* or Dunn's multiple comparisons tests (SPSS or Prism). The detailed information is described in *Supplementary file 1F*. Kolmogorov–Smirnov normality test also was used to determine the application of parametric or nonparametric tests. Statistical significance was set at $p<0.05$. Graphs were made using GraphPad Prism (version 8).

## Data availability

The data that support the findings of this study are available under the project DOI: https://doi.org/105061/dryad2z34tmpn0. The customized analysis tools are deposited on GitHub: https://github.com/KimLab-UW/Crosscorrelation (copy archived at swh:1:rev:f016d43afc82c855ecf-0603ca52a881b9895689e, *Kim, 2019a*) and https://github.com/KimLab-UW/Behavioral_Analysis, (copy archived at swh:1:rev:d866f2f89f53588840d79f4d730f796378e81dad, *Kim, 2019b*).

## Acknowledgements

This study was supported by the National Institutes of Health grant MH099073 (JJK) and the Ministry of Science and ICT through the National Research Foundation of Korea (NRF) grant: Brain Science Research Program NRF-2015M3C7A1028392 (JC), NRF-2019R1A2C2088377 (JC), NRF-2018M3C7A1024736 (YH), and NRF-2020R1A6A1A03043528 (JC).

## Additional information

### Funding

| Funder | Grant reference number | Author |
| --- | --- | --- |
| National Institute of Mental Health | MH088073 | Jeansok John Kim |
| National Research Foundation of Korea | NRF-2015M3C7A1028392 | Jeiwon Cho |
| National Research Foundation of Korea | NRF-2019R1A2C2088377 | Jeiwon Cho |
| National Research Foundation of Korea | NRF-2018M3C7A1024736 | Yeowool Huh |
| National Research Foundation of Korea | NRF-2020R1A6A1A03043528 | Jeiwon Cho |

The funders had no role in study design, data collection and interpretation, or the decision to submit the work for publication.

### Author contributions

Mi-Seon Kong, Conceptualization, Data curation, Formal analysis, Investigation, Project administration, Validation, Visualization, Writing – original draft, Writing – review and editing; Eun Joo Kim, Formal analysis, Methodology, Supervision, Validation, Visualization, Writing – original draft, Writing – review and editing; Sanggeon Park, Formal analysis, Methodology, Resources, Software, Validation; Larry S Zweifel, Methodology, Resources, Supervision; Yeowool Huh, Funding acquisition, Resources, Writing – review and editing; Jeiwon Cho, Formal analysis, Funding acquisition, Methodology, Resources, Supervision, Writing – review and editing; Jeansok J Kim, Conceptualization, Funding acquisition, Supervision, Validation, Writing – original draft, Writing – review and editing

### Author ORCIDs

Mi-Seon Kong http://orcid.org/0000-0001-8970-7034
Eun Joo Kim http://orcid.org/0000-0002-8499-9135
Sanggeon Park http://orcid.org/0000-0003-2083-2536
Larry S Zweifel http://orcid.org/0000-0003-3465-5331
Yeowool Huh http://orcid.org/0000-0002-4652-4059
Jeiwon Cho http://orcid.org/0000-0001-6903-3562
Jeansok J Kim http://orcid.org/0000-0001-7964-106X

### Ethics

All experiments in this study were performed in strict compliance with the University of Washington Institutional Animal Care and Use Committee guidelines (protocol #0404-01). Animals were individually housed in a climate-controlled vivarium (accredited by the Association for Assessment and Accreditation of Laboratory Animal Care) with thorough a daily health checkup. Surgeries were performed under ketamine and xylazine mixture anesthesia to minimize physical discomfort, and post-operative assessments for injury, distress, and pain were followed.

### Decision letter and Author response

Decision letter https://doi.org/10.7554/eLife.72040.sa1
Author response https://doi.org/10.7554/eLife.72040.sa2

## Additional files

### Supplementary files

• Supplementary file 1. *Supplementary file 1A*. The percentage of the basal amygdala (BA) cell types. A total of 250 pyramidal cells were recorded. The bottom table shows the percentage of the cells showing the same, different, or no responses between the pre-robot and the robot sessions. *Supplementary file 1B*. Firing properties of dHPC place cells during the pre-robot, robot, and post-robot sessions. Kruskal–Wallis test (average firing rate, peak rate, spatial info, and running speed) and one-way ANOVA (field size); *p<0.05, **p<0.01, and ***p<0.001 compared to the nest cells; #p<0.05 compared to the proximal cells. *Supplementary file 1C*. Firing properties of place cells during the pre-stimulation, stimulation, and post-stimulation sessions. Data are from the rats with behavioral effects to the photostimulations. Kruskal–Wallis test; **p<0.01 and ***p<0.001 compared to the nest cells; #p<0.05, ##p<0.01, and ###p<0.001 compared to the proximal cells. *Supplementary file 1D*. Firing properties of place cells during the pre-stimulation, stimulation, and post-stimulation sessions. Data are from the rats without behavioral effects to the photostimulations. Kruskal–Wallis test; *p<0.05 and ***p<0.001 compared to the nest cells; #p<0.05 compared to the proximal cells. *Supplementary file 1E*. Descriptions and coordinates for each experiment. *Supplementary file 1F*. Summary of statistical tests performed.

• Transparent reporting form

### Data availability

The data that support the findings of this study are available under the project DOI https://doi.org/10.5061/dryad.2z34tmpn0. The customized analysis tools are deposited on GitHub at https://github.com/KimLab-UW/Behavioral_Analysis (copy archived at swh:1:rev:d866f2f89f53588840d79f4d-730f796378e81dad) and https://github.com/KimLab-UW/Crosscorrelation (copy archived at swh:1:rev:f016d43afc82c855ecf0603ca52a881b9895689e).

The following dataset was generated:

| Author(s) | Year | Dataset title | Dataset URL | Database and Identifier |
|---|---|---|---|---|
| Kong M, Kim E, Park S, Zweifel L, Huh Y, Cho J, Kim J | 2021 | BA-dHPC simultaneous recording | https://doi.org/10.5061/dryad.2z34tmpn0 | Dryad Digital Repository, 10.5061/dryad.2z34tmpn0 |

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

# Appendix 1

## Remapping of distal cells during the predatory encounter

The selective remapping of distal cells during the robot-predator interaction is unlikely due to (i) novelty-related response, (ii) simple sensory stimulus processing, and (iii) motor response-evoked changes. First, the stability of place fields was compared as animals underwent successive pre-robot, stationary robot, looming robot, and post-robot trials (*Appendix 1—figure 1A*). A spatial correlation analysis showed that the mere presence of a novel stationary robot, which elicited no fear in animals, did not disrupt distal place fields. Second, only seven cells (2%, two nest cells and five distal cells) showed significantly increased activities ($Z > 3$) exclusively to the looming robot within 150 ms (*Moita et al., 2003*), and no cells showed a significant increase within 50 ms after the robot activation, which is significantly longer than previously reported sensory-evoked neural responses (<20 ms, *Appendix 1—figure 1B and C*; *Bair et al., 2002*; *Kim et al., 2015*; *Quirk et al., 1997*; *Takakuwa et al., 2018*). In addition, the selective remapping of distal cells still occurred during the first two trials of the post-robot session and the mean foraging time was significantly longer during the first two trials of the post-robot session despite the absence of robot-predator sensory cues (*Appendix 1—figure 1D and E*). As the post-robot trials proceeded, both remapping and foraging time differences gradually disappeared (first three and five trials), and the distal cells became stable comparable to the nest cells. The transient residual effects of the looming robot experience on the stability of place cells and the foraging time during the initial trials of the post-robot session without an explicit threat further indicate that dHPC cell activities and remapping cannot be attributed merely to sensory stimulus processing coupled to the looming robot. Lastly, there were no reliable effects of speed changes on the spatial correlations in the distal cells compared to the nest cells. To determine whether the robot-induced approach/escape behavior or speed changes might have influenced the stability of place cells, we calculated the relative outward between the pre-pellet and pre-surge epochs [(speed$_{pre-surge}$ – speed$_{pre-pellet}$)/(speed$_{pre-surge}$ + speed$_{pre-pellet}$)] and the relative inward speed between the post-pellet and post-surge epochs [(speed$_{post-surge}$ – speed$_{post-pellet}$)/(speed$_{post-surge}$ + speed$_{post-pellet}$); *Kim et al., 2015*]. Then, we analyzed the relationship between the relative speed and the spatial correlations (pre-robot vs. robot sessions, *Appendix 1—figure 1F*). Neither the relative outward nor inward speed was correlated with the spatial correlation in both nest + proximal and distal cells, suggesting that hippocampal remapping during the robot session cannot be explained in entirety by the running speed.

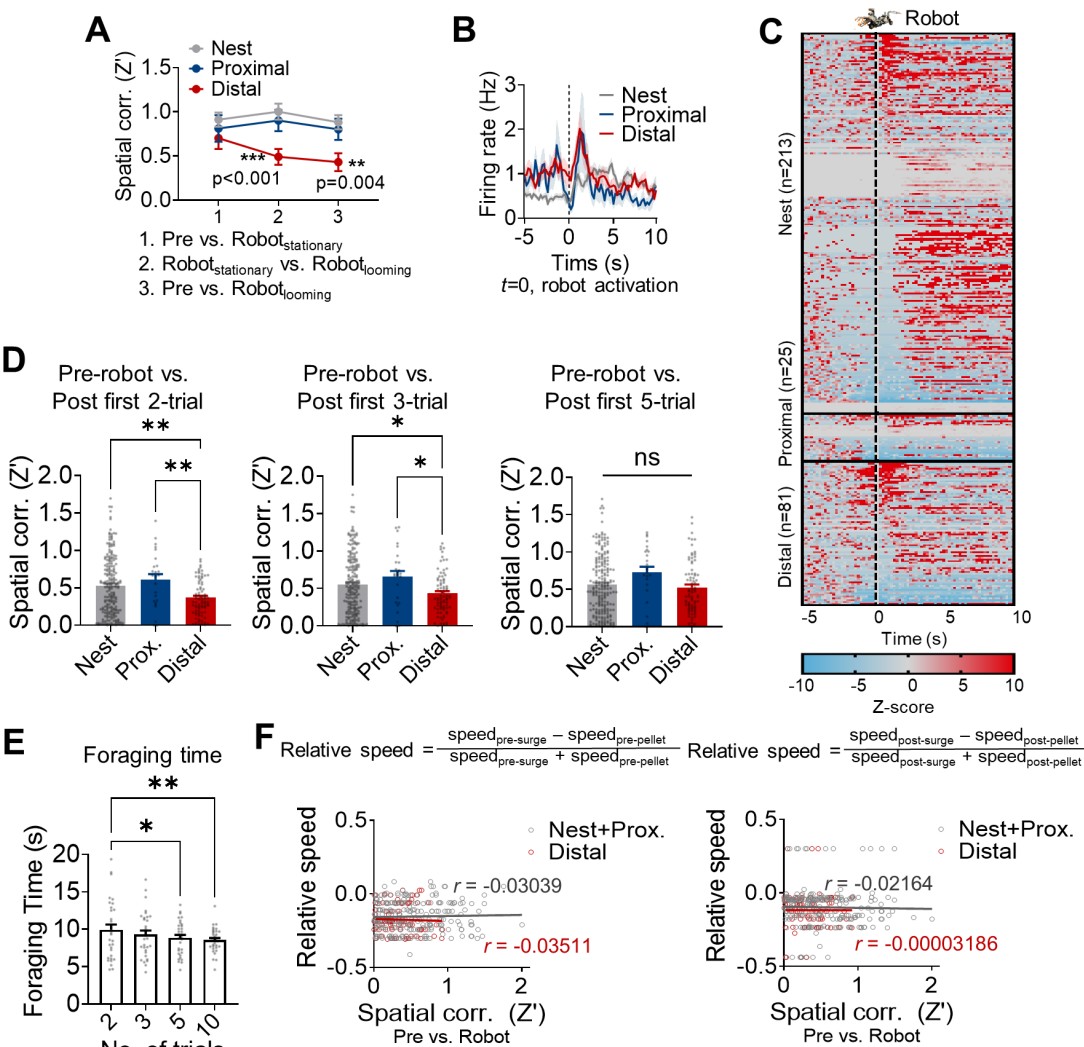

**Appendix 1—figure 1.** The dorsal hippocampus and the risky foraging behaviors. (**A**) Spatial correlations between pre-robot vs. stationary robot (x-axis 1), stationary robot vs. looming robot (x-axis 2), and pre-robot vs. looming robot (x-axis 3) sessions in nest (n = 32), proximal (n = 11) and distal (n = 16) place cells from a total of 6 rats. (**B**) The mean firing rates of the nest, proximal, and distal cells aligned to the robot activation. (**C**) Z-scored activities of all individual dHPC units during –5 s to 10 s after the robot activation (200 ms bin). (**D**) Spatial correlations between the pre-robot vs. first two (left), three (middle), and five (right) trials of post-robot sessions. (**E**) The mean foraging time during the first 2, 3, 5, and 10 trials. (**F**) The correlations between the relative outward (left, pre-pellet vs. pre-surge epochs) or inward (right, post-pellet vs. post-surge epochs) speed of the animals and spatial correlations (pre-robot vs. robot sessions) of the place cells.

