## [Decision Letter]

**Acceptance summary:**

Neurons in the hippocampus which map spatial location (so called 'place cells') are thought to participate in episodic memory encoding and undergo remapping as a result of aversive experiences. The amygdala, an emotional processing center, has been implicated in producing these changes, but the neurophysiological mechanisms through which amygdala neuronal activity produces remapping is not clear. Using dual-site in-vivo electrophysiological recordings, this study found that as animals ventured from a safe location and encountered a simulated predator, activity between amygdala and hippocampal neurons became synchronized and was associated with hippocampal place cell remapping.

**Decision letter after peer review:**

[Editors’ note: the authors submitted for reconsideration following the decision after peer review. What follows is the decision letter after the first round of review.]

Thank you for submitting your work entitled "'Fearful-place' coding in the amygdala-hippocampal network" for consideration by *eLife*. Your article has been reviewed by 3 peer reviewers, one of whom is a member of our Board of Reviewing Editors, and the evaluation has been overseen by a Senior Editor. The reviewers have opted to remain anonymous.

We are sorry to say that, after consultation with the reviewers, we have decided that your work will not be considered further for publication by *eLife* at this time.

While the reviewers found the idea of interactions between specific populations of amygdala and hippocampal neurons related to hippocampal place cell remapping potentially interesting, there was a consensus opinion that the conclusions drawn from these studies were not justified given the current state of the data analysis. There were also concerns about the novelty of some of the results as similar findings have been reported in previous papers from your group. Based on these and other considerations (see detailed reviews below) and the fact that *eLife* aims to publish papers with a single round of revision that under normal circumstances can be accomplished within two months, we decided that your paper cannot be considered further at this time.

Should further experimental analyses or data allow you to address the issues raised by the reviewers we could consider a resubmission. Please refer to this manuscript number if you choose this route. We cannot make any guarantees about a resubmission at this point, however. We would need to be convinced that the reviewer concerns had been addressed before committing to a re-review. We do not intend any criticism of the quality of the data or the rigor of the science. We wish you good luck with your work and we hope you will consider *eLife* for future submissions.

*Reviewer #1:*

Kong and colleagues in Jeansok Kim's lab studied how place cells in the in dorsal hippocampus (dHPC) coordinate with neurons in the basal nucleus of the amygdala (BA) during exposure to a frightening robot-alligator pseudopredator to produce remapping of hippocampal place cell representations. Prior work has demonstrated place cell remapping upon exposure to aversive events. The Kim lab has extended this by showing that exposure to the pseudopredator they use in the present paper also causes dorsal dHPC place cell remapping and increased theta power, that these changes are not apparent when the amygdala is lesioned and that electrical stimulation of the amygdala produces CA1 remapping and avoidance behaviors. In the current paper they replicate their previous results on pseudopredator induced remapping and increased theta power in dHPC cells and stimulation (optogenetic in this case) induced remapping of dHPC cells. In addition, they show that dHPC cells that are closest to the predator and whose firing is coordinated with BA neurons which respond to predator exposure.

While some of this work is a replication of their previous findings and undermines the novelty of the current paper a bit, the possible demonstration of how BA-dHPC cells are coordinated to produce hippocampal place cell remapping during predator experiences is interesting and potentially important. However, there are a number of interpretational issues with the results which make it difficult to determine whether the conclusions are supported by the data. Specifically, it is not clear whether this is in fact destabilization of place cell firing or a switch to predator driven responsiveness in dHPC cells. It is also unclear whether the apparent remapping in dHPC cells is in fact neuronal activity in BA cells or simply a consequence of cells in both regions being co-responsive to the predator stimulus.

1) The authors show that place field stability is lower when the robogator is present/attacking, but I'm not sure this can be called remapping. The fact that there is a different sensory environment with the presence of the robogator and that it is attacking could produce what looks like remapping but is in fact a sensory (or aversive) response to the robogator. These possibilities are supported by the fact that place fields are somewhat stable when comparing pre to post robot (Figure 1-Suppl 1J-M), suggesting that this is not remapping which would likely be expressed in a more long term way (e.g. at the 'post-robot' timepoint). These concerns cannot be ruled out by the fact that amygdala lesions reduce escape from predator behaviors and block place cell changes (Kim et al.et al., 2015), as the authors suggest in the Discussion (pg. 15), because novelty, sensory, aversive or motor information are encoded in and could be conveyed from the amygdala.

2) Because the rats did not spend much/any time venturing into the robot adjacent portion of the chamber, it is possible that there was an impairment in the author's ability to map and compare place fields during the Robot portion of the experiment to pre and post Robot. For example, if the proximal place cell representation in some cells were close to the Robot, but they only ventured to the edge of the place field, that could look like a place field shift.

3) Directly related to point 1, the increase in spike synchrony in BA-HPC cells could be related to an increase in firing rates (possibly induced by the predator event) in BA and HPC cells (co-modulation) and not to the directional influence of one on the other.

4) Also related to the above points, the selective degradation of place cell stability in dHPC cells and the significant cross-correlations with BA neurons in these cells could result from a coordinated increase in firing rates during/after robogator experience in these cell pairs. In this scenario, both remapping and correlated spiking are secondary to coordinated increases in firing rate in what, on the surface, appear to be 'connected' pairs.

5) Point 3 also applies to the optogenetic stimulation effects on dHPC place cell stability shown in Figure 4. The BA stimulation could be increasing firing rates in some of the dHPC cells resulting in a change in the spatial tuning of the cells.

6) Another more minor issue is that a majority of the prior literature has not detected any major input from the BA to the dorsal hippocampus but do see strong inputs to the ventral hippocampus. The Petrovich, 2001 and Pitkanen, 2000 papers the authors cite that examined the BA-hippocampus connectivity systematically also don't see a major projection to dCA1, although the Rei paper does suggest that this connection exists. If the direct connections are minor, this doesn't seriously undermine the overall conclusions of the study, but should be acknowledged because the combined effect of the tracing and cross-correlation results in the current version of the paper imply a monosynaptic BA-dHPC connectivity.

7) The authors show that place field stability is lower when the robogator is present/attacking, but I'm not sure this can be called remapping. The fact that there is a different sensory environment with the presence of the robogator and that it is attacking could produce what looks like remapping but is in fact a sensory (or aversive) response to the robogator. These possibilities are supported by the fact that place fields are somewhat stable when comparing pre to post robot (Figure 1-Suppl 1J-M), suggesting that this is not remapping which would likely be expressed in a more long-term way (e.g. at the 'post-robot' timepoint). These concerns cannot be ruled out by the fact that amygdala lesions reduce escape from predator behaviors and block place cell changes (Kim et al.et al., 2015), as the authors suggest in the Discussion (pg. 15), because novelty, sensory, aversive or motor information are encoded in and could be conveyed from the amygdala.

Suggestions for improvement: To approach the sensory questions they could examine how CA1 cells respond to the robot attack itself, as they did with the BA cells in Figure 1E-F (e.g. include heat plots and population averaged peri-event histograms), though this does not deal with the problem completely. They could also examine whether in early trials animals of 'post robot' animals still exhibit avoidance behavior in the absence of the robot and whether there is still remapping compared to 'pre-robot'. In addition, they could change the framing of the paper and discuss how this could be a sensory related response and not necessarily a remapping event.

8) Because the rats did not spend much/any time venturing into the robot adjacent portion of the chamber, it is possible that there was an impairment in the author's ability to map and compare place fields during the Robot portion of the experiment to pre and post Robot. For example, if the proximal place cell representation in some cells were close to the Robot, but they only ventured to the edge of the place field, that could look like a place field shift.

9) Directly related to point 1, the increase in spike synchrony in BA-HPC cells could be related to an increase in firing rates (possibly induced by the predator event) in BA and HPC cells (co-modulation) and not to the directional influence of one on the other.

Suggestions for improvement: The authors could address this by examining whether there was a tendency in the cell pairs across the population to show an increase in firing rate during pre/post surge periods and examine whether the average lag time between BA and dHPC spikes were around 0, shifted to negative or positive or was bimodally distributed across the cells (possibly using a frequency histogram showing cell counts using the lag times for each cell). They could also examine whether increases in firing rate covaried or were independent from the cross-correlation strength on individual trials or whether significant cell pairs in one condition lost their correlation in another condition where they still showed enhanced firing rates.

10) Also related to the above points, the selective degradation of place cell stability in dHPC cells and the significant cross-correlations with BA neurons in these cells could result from a coordinated increase in firing rates during/after robogator experience in these cell pairs. In this scenario, both remapping and correlated spiking are secondary to coordinated increases in firing rate in what, on the surface, appear to be 'connected' pairs.

11) Point 3 also applies to the optogenetic stimulation effects on dHPC place cell stability shown in Figure 4. The BA stimulation could be increasing firing rates in some of the dHPC cells resulting in a change in the spatial tuning of the cells.

12) Another more minor issue is that a majority of the prior literature has not detected any major input from the BA to the dorsal hippocampus but do see strong inputs to the ventral hippocampus. The Petrovich, 2001 and Pitkanen, 2000 papers the authors cite that examined the BA-hippocampus connectivity systematically also don't see a major projection to dCA1, although the Rei paper does suggest that this connection exists. If the direct connections are minor, this doesn't seriously undermine the overall conclusions of the study, but should be acknowledged because the combined effect of the tracing and cross-correlation results in the current version of the paper imply a monosynaptic BA-dHPC connectivity.

Suggestions for improvement: The authors should more carefully describe their anatomical findings including from Figure 2-Suppl Figure 2A and show a) broader images in which hippocampal subareas can be seen, b) quantification of the cell number c) think about performing an anterograde tracing experiment looking at terminal innervation of CA1 from BA tracer injections. They could also look at the latency of the cross-correlation values in individual cells as well as the latency of the optogenetic stimulation induced increases in firing rate to address this issue. If direct projections are minor, the authors should consider the possibility that the cross-correlations occur through indirect connectivity (which they do to some extent in the Discussion). If so, this doesn't seriously undermine the overall conclusions of the study, but should be acknowledged because the combined effect of the tracing and cross-correlation results in the current version imply a monosynaptic BA-dHPC connectivity.

*Reviewer #2:*

In this paper the authors aim at identifying and testing reciprocal influence of the space and threat representations mediated by the dHPC and BA. To do so, they have recorded neuronal activity simultaneously in the dorsal hippocampus (dHPC) and basal amygdala, in a naturalistic foraging task with a looming robot simulating a threatening predator. In this task the animal venture from the nest into a foraging zone to retrieve a food pellet. In one of the sessions, the robot is activated as soon as the animal approaches the pellet, triggering a flight-type reaction (returning to the nest). The authors identify different subsets of BA neurons : threat-responsive and pellet-responsive, and then focus their analysis on threat responsive BA neurons (robot-cells) and study their correlation with the dorsal hippocampus neurons. They then use optogenetics to stimulate BA during foraging, mimicking the BA activation by the robot, and assess the effects of this stimulation on behavior and dHPC place cell stability.

The difficulty for assessing place cell activity in threatening situations is that the reaction triggered, in this case hesitation and pausing, constitute strong modification of the exploratory behavior usually requested to analyze place cell activity. It is then hard to pinpoint what is due to actual place-cell property modification and what is due to differences in behavior. The authors have addressed this question in previous studies (not reviewed here). In this paper they reproduce the same results as presented before and add an incremental step by using optogenetics to activate BA and trigger place field instability (they previously established that BA inactivation or lesion prevents place field disruption). A strength of the paper is that optogenetic stimulations of the BA, although triggering the same flight response as the robot, does not seem to trigger the same pausing and hesitations in the approach phase, therefore making this part less prone to spurious results in remapping due to behavioral changes.

The major interest and novelty in this study, in my opinion, is the use of this valuable dataset (dHPC and BA simultaneous recordings in a naturalistic foraging task) to look at the fine temporal dynamics between BA and dHPC. Unfortunately, this is also very difficult due to the very small number of correlated cell pairs. Indeed, despite the authors's tracing data and previous literature on it, the presence of direct connections between dHPC and BA is not fully established and, at best, very sparse. This sparsity, or the possibility of correlations through other structures, doesn't make this study less interesting, only more difficult. Indeed, although promising, the author's claims that the BA and HPC reciprocally interact on a short time window around the robot "surge" or closer to the threat is only partially backed by their results that require, at this stage, more careful statistical controls.

– It seems that figures G and H are essentially recapitulating previous work done by the authors (Kim et al.et al. 2015). In particular, how is Figure 1H in the present paper different from figure 2c in the 2015 one?

– Remapping measures : the peak distance measure isn't clear and is not described in the methods. In the text it is mentioned pauses in the foraging behavior in the robot session. However average running speeds are higher, which I suppose is due to faster running/escaping to the nest. This means there are major speed/behavior differences in the robot and non-robot epochs that could be described/accounted for. In the same line of thought, I suggest an additional criteria for place cells of minimal velocity. This would be ensuring that the remapping is not due to behavioral differences between pre-robot and robot sessions.

– Since the BA is supposed to encode both positive and negative valences, the "pellet-cells" should be included in the analysis, represented as in figure 1D and shown in figure 1F.

– If I understood the methods correctly, the data are normalized separately for the pre-robot and robot-sessions. Is it correct then to substract the z-values (figure 1f) coming from two different normalizations? (+ the number of cells in the text and figure do not match).

– I would include a pie chart to clearly state robot, pellet, pellet-robot and neutral cells percentages and numbers in the principal figure.

– CTB tracking measures : there is only one example, and the number of animals is not mentioned. There is no quantification. The cell bodies in CA1are blurry and since we don't see the rest of the picture this could well be baseline fluorescence. From the picture if we trust it it also seems that CA1 to BA projecting cells are much more numerous than BA to CA1 projecting cells. This supplementary figure is not convincing to me. I think it should be properly performed or removed. The paper does not need to establish direct connections between the dHPC and the BA to be relevant, provided the results are commented appropriately. Properly establishing this with tracing methods coupled with physiology (ie monosynaptic connections on cross-correlograms) is a whole other endeavor.

– The analyses presented in figure 2 are the one that require more careful control. First, I do not understand why the authors separated pre-surge and post-surge epochs. Then, the 100ms window, although justified by the authors from connectivity data, seem at least partially arbitrary (especially since they also show a long decay -10s- in figure 1F). Finally, restricting the analysis to such little data (the number of surge events/total recording time on which the ccgs are calculated should be stated) induces baselines close to zero and makes it vulnerable to spurious correlations. Then, figures 2C and F illustrate a circular reasoning because the authors are showing significant differences on the averages of cells that were selected based on this difference (during the window).

My suggestions are to : – put a limit of a minimal number of spikes in the considered windows (rather than an overall minimal firing rate) – perform the same analysis on randomly chosen 100 windows to evaluate the number of cell pairs that would be identified by chance (or potentially come up with other shuffling strategies). – Vary the length of the chosen window and see if the results are consistent.

*Reviewer #3:*

The authors investigate the impact of amygdala activity on coding of place in the hippocampus. They use a paradigm pioneered and established by this group in the past several years, which nicely combines foraging for food with predatory risk/fear. Rats were required to go out of their nest to look for food pellets while in an environment that includes a "scary" robot, designed to mimic predatory risk. This group has previously shown by lesions, inactivation, and disinhibition, that the amygdala regulates risk behavior (fear of the predator-robot) while foraging for food. Additionally, in a follow-up study, they recorded hippocampal place-cells and showed that spatial information is altered ("remapping") by foraging in the robot risky environment and that amygdala lesions prevent this remapping. In the prevent study they seek to establish this framework even further, and they therefore use two strategies: 1. Simultaneous recordings from the amygdala and the hippocampus that allow cross-correlations between single-units in both structures; and 2. Optogenetic excitation of amygdala cells while examining place coding and remapping in the hippocampus. Both approaches provide further evidence to the previous published findings and are overall well performed and reported with clarity. The conclusions are reasonably supported by the data, but the novelty is less clear, and the novel data does not provide clear insights.

The strengths of the paper are the simultaneous recordings of spikes in the amygdala and the hippocampus, which allow more direct evaluation of the effect of amygdala activity on remapping of hippocampal place-cells. The main finding is that remapping in the hippocampus occurred mainly at hippocampal cells that showed synchronous activity with robot-responsive amygdala cells. The other strength could be the optogenetic activation of amygdala cells and the effect on behavior and hippocampal remapping, providing a more concrete evidence that it is indeed increased activity in the amygdala that regulates behavior and place coding under threat.

The main weakness is that the results do not provide substantial additional insights beyond the already published studies.

The cross-correlations are a useful approach, but it is not clear how many of the overall CCs (1999) are independent, namely result from different pairs (vs how many include the same individual neuron). Adding to this the finding that the number of significant CCs is not really high and fluctuates around chance levels (5% or so), makes the finding hard to interpret. Additionally, "non-robot" BA cells are defined as cells that did not respond to any event (robot or pellet), and these are therefore cells which potentially do not have any interest in the paradigm and of lower activity overall. As a result, they do not provide a proper control for CCs that might be related to "remapping" of hippocampal place cells. Finally, the correlation between position along the arena and the "remapping" are very nice but might well be a result of two groups (nest/proximal, and distal), rather than a real gradual correlation. This needs to be analyzed more carefully.

The optogenetics is a strength in principle but limited as well. First, if the authors would like to claim that it is indeed BA->HPC, then the power of optogenetics should be used by injecting in one place and stimulating in the other. Second, a control region should be used in addition to the amygdala, to make sure that the stimulation itself (cell excitation) is not in itself a fearful experience for the animal.

Finally, the "remapping" is the finding that spatial sensitivity has changed, but there is no constructive value in it. In other words, if a place has a predator associated with it, then cells should either represent this location better (to remember and to avoid it), or they should remap to code for something else (e.g. a place that has other pellets and no risk). While one cannot ask the data to show something if it does not, it currently does not provide much beyond the previous studies, and further analyses are required to help understand how remapping is modulated by the amygdala to enable future behavior (e.g. in the post-robot period).

In addition to what was mentioned in the previous section:

1. Some controls are required to make sure that place-coding was not disrupted (remapping) by differences in visiting the distal regions due to the robot risk (different trajectories, less sampling as evident by low success rate etc.).

2. The division between proximal to distal should be uniform, either in distance and/or in number of place-cells, both are required for correct interpretation.

3. Remapping (and also BA activity could result from distance from the nest (less safety), and this has been shown in many studies. Currently there is no way to dissociate the effect of the robot from the effect of the distance from the nest (they complement each other). Because all sessions are the same: pre-robot, robot, and post-robot, and there are not specific differences between cells in robot and post-robot, it is hard to dissociate the two factors. A future study can include sessions when a robot surges in a proximal location.

4. Another way to think about the previous concern, is to notice that by definition, a robot cell is a distal cell in the amygdala, and therefore CCs are expected (as nicely exemplified in the scheme in figure 5). a better design can dissociate the two factors.

5. The findings of CCs, central to the study, are not reported in a clear manner and one has to follow the numbers and the segmentation very carefully.

6. It is not clear when exactly photostimulation was delivered during the session.

7. More "constructive" characterization of the remapping would really improve the study and its novelty. Do cells remap to code for a different place? How is it in relation to the pellet locations? Why should place-coding be disrupted if the animal wants to remember the location of a threat ?

[Editors’ note: further revisions were suggested prior to acceptance, as described below.]

Thank you for resubmitting your work entitled "'Fearful-place' coding in the amygdala-hippocampal network" for further consideration by *eLife*. Your revised article has been evaluated by Laura Colgin (Senior Editor) and a Reviewing Editor.

The manuscript has been improved but there are some remaining issues that need to be addressed, as outlined below. Specifically, further analyses requested by Reviewers 1 and 2 are required to shore up the findings. Furthermore, the addition of some type of off-site control stimulation group for the optogenetic study would strengthen the paper but including a further discussion of the caveats/potential limitations of the single-site stimulation approach may be sufficient.

Essential revisions:

*Reviewer #1:*

The authors have addressed most of my previous concerns and focused the paper around the more novel aspects of the findings related to the role of spiking synchronization between basal amygdala (BA) and dorsal hippocampus (dHPC) neurons during aversive experiences and the role of this process in producing experience dependent remapping of dHPC cells. This remapping is maintained transiently after the predator is removed but while animals are still exhibiting avoidance behaviors and not correlated with changes in firing rate, concerns from past reviews. Notably, BA optogenetic stimulation alone is sufficient to reduce foraging behavior and induce dHPC cell remapping. This provides a physiological mechanism for the authors' previous finding that BA lesions block predator induced dHPC cell remapping.

While these results are interesting, there is one remaining issue which should be addressed: Related to Figure 1 showing BA-dHPC synchrony, the authors report the percentage of cells showing synchrony around the pre/post predator encounter and they show that this did not occur in these aversive cell pairs during a non-aversive experiences (i.e. pre/post pellet procurement). However, they did not report any data on the pre/post pellet encounter, explicitly examining the percentage of cells showing significant cross-correlations (CCs) during these periods. This should be included along with an analysis of the overlap between cells with significant CCs during aversive or rewarding experiences if there is a large population of cells showing this for the pellet encounters. This will give the reader a better understanding of the nature of this synchronization, whether it is related specifically to aversive encounters and, if so, whether it occurs in separate populations of cells during these different types of experiences.

*Reviewer #2:*

The authors did a fine job addressing most of the concerns, and performed additional analyses producing new figures and results.

Yet, I still feel that the main concern: "The main weakness is that the results do not provide substantial additional insights beyond the already published studies.", did not receive a direct and appropriate response in the letter. The CC and the optogenetics are an obvious novelty on one hand, yet on the other, the finding that BA activity in fearful environments is a pre-cursor for remapping of a [any] behavioral correlate that is relevant for the task, is not really novel. If the details and the mechanisms were more convincing, revealing something new about BA-HPC interactions, or about the remapping itself, then it could have been a more interesting study. Overall, I do not have a strong objection to the work, but I admit I am not convinced that it will make a major impact beyond previous work.

In addition, some concerns were not fully addressed. For example:

The optogenetics is a strength in principle, but limited as well. First, if the authors would like to claim that it is indeed BA->HPC, then the power of optogenetics should be used by injecting in one place and stimulating in the other. Second, a control region should be used in addition to the amygdala, to make sure that the stimulation itself (cell excitation) is not in itself a fearful experience for the animal.

Their answer: We agree with the first comment if BA neurons supposedly influence dHPC neuronal activities via their monosynaptic projections. However, as originally stated in the manuscript, the majority of amygdala and dHPC projections is polysynaptic (via dorsal CA3 and ventral HPC areas). Given the sparsity of direct BA-dHPC connection and other reviewers' recommendations (i.e., Reviewer 1's comment #6 and Reviewer 2's comment #6), the CTB tracing data are now removed.

This is an argumentative response. If we generalize this type of response, then why do we need any kind of study that injects in one place and stimulate in the other? There are polysynaptic connections between almost any two region in the brain, hence stimulating one does not demonstrate conclusively that changes that occur in the other are caused by it. Perhaps the BA responses are what they are: BA responses to a fearful environment (as we know fo many years), and HPC remapping occurs due to other inputs from other region independent of the BA? I admit the CCs and the general common sense (hypothesis) in the field makes this less likely, but that is exactly my point about the novelty. Either you show the BA-HPC in a more convincing way, or it remains a correlational nice study with limited novelty.

Their answer: As for the second comment, we do not have a different brain region stimulation 'control' group, which can potentially raise further questions because stimulating a different region can affect/alter other (e.g., nondefensive) behaviors and neural circuits. However, we do have three control conditions, (i) EYFP control (the traditional channelrhodopsin control), (ii) low laser power control, and (iii) optic fiber misplacement control. In all three control conditions, we did not detect defensive behaviors at the timing of light delivery, and in the case of ii and iii control groups, their place cells showed no remapping. These findings suggest that the BA stimulation per se (in the absence of eliciting defensive responses) is not sufficient to cause place cell remapping.

I do not understand why they did not perform few additional experiments with this control?

Their claim that "stimulating a different region can affect/alter other (e.g., nondefensive) behaviors and neural circuits" is directly what they would want to test – that it is indeed BA inputs per-se, and not other changes. Moreover, their response "BA stimulation per se (in the absence of eliciting defensive responses)" indeed raises the possibility that BA stimulation induced a fearful state not because it was in the BA, but because the animal felt something unknown (i and ii do not address this) inducing a fearful state, and hence remapping (and again we are left without knowing if it is via the BA).

To emphasize again: I have no doubt that their hypothesis is likely correct, but the study as-is does not tell us much more than we already knew.

1. Some controls are required to make sure that place-coding was not disrupted (remapping) by differences in visiting the distal regions due to the robot risk (different trajectories, less sampling as evident by low success rate etc.).

Their answer: The original manuscript explicitly addressed this concern (pg. 5): "Because the looming robot prevented the animals from reaching the pellet location, all neural analyses was based on equating the nest-to-foraging distance in each trial of pre-robot, robot and post-robot sessions (Figure 1A)." However, to better clarify this, we added a note…

This does not answer the main concern. It does address the analyses, but it does not address the potential confound those different trajectories and frequency of visiting distal regions contributed to the remapping. Namely that behaviors that are "outside" of the time/place taken for analyses, took part in remapping. Perhaps I am missing something in their controls?

2. The division between proximal to distal should be uniform, either in distance and/or in number of place-cells, both are required for correct interpretation.

In the previous place cell recording study (Kim et al.et al., 2015), we defined the proximal region as the foraging area between 0-25 cm from the nest since rats failed to procure the pellet beyond this proximal-distal boundary when facing a looming robot (Choi and Kim, 2010). We have also reported that when the distal zone was subdivided into two areas (one relatively nearer to the nest and other farther from the nest), there was no reliable difference in spatial correlation and the peak distance between the two distal areas.

This is great. Why not repeat the same analyses here and show it is similar? One cannot rely on a previous study for such controls.

---

## [Author Response]

[Editors’ note: the authors resubmitted a revised version of the paper for consideration. What follows is the authors’ response to the first round of review.]

Reviewer #1:Kong and colleagues in Jeansok Kim's lab studied how place cells in the in dorsal hippocampus (dHPC) coordinate with neurons in the basal nucleus of the amygdala (BA) during exposure to a frightening robot-alligator pseudopredator to produce remapping of hippocampal place cell representations. Prior work has demonstrated place cell remapping upon exposure to aversive events. The Kim lab has extended this by showing that exposure to the pseudopredator they use in the present paper also causes dorsal dHPC place cell remapping and increased theta power, that these changes are not apparent when the amygdala is lesioned and that electrical stimulation of the amygdala produces CA1 remapping and avoidance behaviors. In the current paper they replicate their previous results on pseudopredator induced remapping and increased theta power in dHPC cells and stimulation (optogenetic in this case) induced remapping of dHPC cells. In addition, they show that dHPC cells that are closest to the predator and whose firing is coordinated with BA neurons which respond to predator exposure.While some of this work is a replication of their previous findings and undermines the novelty of the current paper a bit, the possible demonstration of how BA-dHPC cells are coordinated to produce hippocampal place cell remapping during predator experiences is interesting and potentially important. However, there are a number of interpretational issues with the results which make it difficult to determine whether the conclusions are supported by the data. Specifically, it is not clear whether this is in fact destabilization of place cell firing or a switch to predator driven responsiveness in dHPC cells. It is also unclear whether the apparent remapping in dHPC cells is in fact neuronal activity in BA cells or simply a consequence of cells in both regions being co-responsive to the predator stimulus.1) The authors show that place field stability is lower when the robogator is present/attacking, but I'm not sure this can be called remapping. The fact that there is a different sensory environment with the presence of the robogator and that it is attacking could produce what looks like remapping but is in fact a sensory (or aversive) response to the robogator. These possibilities are supported by the fact that place fields are somewhat stable when comparing pre to post robot (Figure 1-Suppl 1J-M), suggesting that this is not remapping which would likely be expressed in a more long term way (e.g. at the 'post-robot' timepoint). These concerns cannot be ruled out by the fact that amygdala lesions reduce escape from predator behaviors and block place cell changes (Kim et al.et al., 2015), as the authors suggest in the Discussion (pg. 15), because novelty, sensory, aversive or motor information are encoded in and could be conveyed from the amygdala.Suggestions for improvement: To approach the sensory questions they could examine how CA1 cells respond to the robot attack itself, as they did with the BA cells in Figure 1E-F (e.g. include heat plots and population averaged peri-event histograms), though this does not deal with the problem completely. They could also examine whether in early trials animals of 'post robot' animals still exhibit avoidance behavior in the absence of the robot and whether there is still remapping compared to 'pre-robot'. In addition, they could change the framing of the paper and discuss how this could be a sensory related response and not necessarily a remapping event.

As recommended by the reviewer, we aligned dHPC activities to the robot activation to address whether the decreased stability of distal place fields reflects direct sensory responses to the Robogator as opposed to Robogator-induced fear. Appendix—figure 1B and C, shows the nest, proximal and distal cells’ mean firing rate changes to the looming robot. All three cell types increased their activities after the robot activation, but their peak-firing appeared > 1 second after the robot activation, which is significantly longer than previously reported < 20 ms sensory-evoked neural responses (cf. Bair et al.et al., 2002; Quirk et al.et al., 1997; Takakuwa et al.et al., 2018). Of all individual cell responses are plotted in Appendix—figure 1C, only 7 cells (2.7%, 2 nest cells, 5 distal cells) showed significantly increased activities (Z > 3) exclusively to the looming robot within 150 ms (cf. Moita et al.et al., 2003), and no cells showed significant increase within 50 ms after the robot activation. Given the delayed responses, the dHPC cell activities and remapping are likely driven by affective (i.e., fear) states, rather than sensory stimulus (i.e., robot) processing coupled to the surging robot per se.

Another valuable recommendation by the reviewer was to examine whether remapping occurred during the early trials of the post-robot session. We thus calculated spatial correlations between the pre-robot session and the early trials of post-robot session (i.e., first two, three and five trials). As can be seen in Appendix—figure 1D, the spatial correlations of distal cells between prerobot and first two trials of the post-robot sessions were significantly lower compared to those of nest and proximal cells. As the trials proceeded, the significant difference disappeared (first three and five trials) and the distal cells became stable when compared to the nest cells (Appendix—figure 1D). Consistent with this, the mean foraging time to successfully procure the pellet during the first two trials was significantly longer compared to the latter trials (Appendix—figure 1E). These results indicate that there were transient residual effects of the robot experience on the stability of place cells and the foraging time during the initial trials of the post-robot session when an external threat stimulus was absent. However, the selective remapping of the distal cells and increased foraging time quickly reverted to the baseline levels during the remaining trials of the post-robot session.

Overall, we believe the use of the term ‘remapping’ to describe our findings of distal (but not nest-proximal) place fields in a familiar environment shifting to unpredictable locations with Robogator-induced fear in the same environment have been strengthened by presenting that (i) dHPC place cell activity to the robot activation occurred outside the range of direct sensory responsiveness, and (ii) distal cells continued to display residual remapping without the looming Robogator during the initial trials of the post-robot session. The reviewer noted that novelty and sensory effects on the distal cell remapping cannot be entirely ruled out by our previous findings of amygdala lesions reducing both escape behavior and place cell alterations (Kim et al.et al., 2015). The same study, however, also showed distal cell remapping only in animals that failed to procure pellets (high-fear state) but not in those that sporadically showed successful foraging (low-fear state) during the robot session. Moreover, there is evidence of amygdala-lesioned rats exhibiting normal novel object recognition memory (Aggleton et al.et al., 1989; Mumby and Pinel, 1994) and displaying normal learning and enhanced memory of a visually salient platform water maze task (Kim et al.et al., 2001). Taken together, the present findings are consistent with the notion that fear elicited by the looming Robogator (and not Robogator’s sensory features) is the likely factor of distal cell remapping.

2) Because the rats did not spend much/any time venturing into the robot adjacent portion of the chamber, it is possible that there was an impairment in the author's ability to map and compare place fields during the Robot portion of the experiment to pre and post Robot. For example, if the proximal place cell representation in some cells were close to the Robot, but they only ventured to the edge of the place field, that could look like a place field shift.

The original manuscript explicitly addressed this concern: “Because the looming robot prevented the animals from reaching the pellet location, all neural analyses were based on equating the nest-to-foraging distance in each trial of pre-robot, robot and post-robot sessions (Figure 1A).” The revised manuscript better clarified this by appending a note on Figure 1A and by editing the above statement to “Because the looming robot prevented the animals from reaching the pellet location, the distal cells that had place fields beyond the foraging limit (where the animal did not visit during the robot session) were excluded for any place cell analyses to equate the nest-to-foraging distance throughout the sessions.”

3) Directly related to point 1, the increase in spike synchrony in BA-HPC cells could be related to an increase in firing rates (possibly induced by the predator event) in BA and HPC cells (co-modulation) and not to the directional influence of one on the other.Suggestions for improvement: The authors could address this by examining whether there was a tendency in the cell pairs across the population to show an increase in firing rate during pre/post surge periods and examine whether the average lag time between BA and dHPC spikes were around 0, shifted to negative or positive or was bimodally distributed across the cells (possibly using a frequency histogram showing cell counts using the lag times for each cell). They could also examine whether increases in firing rate covaried or were independent from the cross-correlation strength on individual trials or whether significant cell pairs in one condition lost their correlation in another condition where they still showed enhanced firing rates.

First, all cross-correlations data presented in the original manuscript were corrected by ‘Shift-Predictor,’ ’ where 100 times of trial shuffles were applied. Each shift predictor correlogram was subtracted from its respective raw correlogram, and Z-scores were calculated by the mean and standard deviation of the corrected CC. The neural pair was considered to be significantly correlated if the peak Z-score was > 3. Additional criteria were that the BA and dHPC firing rates during the pre- and post-surge periods must be above 0.1 Hz, and the peak of the CC should fall within a testing window of ±100 ms relative to the reference spikes. This detailed information has been updated in the Materials and methods section (Cross-correlation). We believe that these correction and strict criteria for determining significant cross-correlations sufficiently exclude the chance of false correlations due to covariation or nonstationary firing rates from the BA and dHPC.

To further exclude the likelihood of ‘co-firing modulation effects on cross-correlations and spatial correlations’, we present additional analyses showing that (i) during the pre-surge, both BA and dHPC cells did not display time-locked responses to the specific event, and the response latencies (peak responses) following the robot surge were not overlapped between the BA and dHPC cells (Figure 1F,H in the revised manuscript), (ii) while BA Robot cells increased firing compared to nonRobot cells (Figure 2A and Figure 2─figure supplement 4C in the revised manuscript), there were no reliable differences between robot cell-paired distal cells and nest+proximal cells in their firing rates either during the pre-surge or post-surge period (Author response image 1 and Figure 2I in the revised manuscript), and (iii) that dHPC firing rates comparably increased in all nest, proximal and distal foraging regions (Appendix—figure 1B) yet all possible pairs (Robot/nonRobot-nest/proximal/distal) showed spike synchrony with lower proportions of Robot-place cell pairs; (31%, pre-surge; 34%, postsurge; Figure 2─figure supplement 3B in the revised manuscript), and (iv) only the Robot-paired distal cells (but not nonRobot-paired distal or Robot/nonRobot-paired nest+proximal cells) showed decreased spatial correlation between the pre-robot and robot sessions (Figure 2G,H in the revised manuscript). These results suggest that the increase in spike synchrony in BA-dHPC cells cannot fully be accounted by increased firing rates.

**Author response image 1. sa2fig1:** The mean firing rates during the pre-surge (left) and post-surge (right) epochs of Robot/nonRobot cell-paired distal/nest+proximal cells.

As recommended, we also looked at ‘Robot+Pellet BA-dHPC’ cell pairs (Figure 2─figure supplement 4). Those pairs that showed significant synchrony during the pre-surge epoch also showed comparable firing rates during the pre-pellet epoch (2.5 s before the pellet acquirements during the pre-robot session). The mean firing rates between the pre-pellet and pre-surge epochs were not different (Figure 2—figure supplement 4E)*,* yet their cross-correlations (CCs) were significant only during the presurge epoch (shown as the peak Z-scored dHPC spikes at the time of BA spikes during each epoch, Figure 2—figure supplement 4F). The same results were obtained for the pairs that showed significant synchrony during the post-surge epoch. Their firing rates were not different between the post-pellet (2.5 s after the pellet acquirements during the pre-robot session) and the post-surge epochs, but CCs were only significant during the post-surge epoch (Figure 2—figure supplement 4G,H). These results again confirm that co-firing between the two areas does not always lead to ‘synchronous firing’.

4) Also related to the above points, the selective degradation of place cell stability in dHPC cells and the significant cross-correlations with BA neurons in these cells could result from a coordinated increase in firing rates during/after robogator experience in these cell pairs. In this scenario, both remapping and correlated spiking are secondary to coordinated increases in firing rate in what, on the surface, appear to be 'connected' pairs.

In our reply to comment #3, we showed that BA Robot cells increased firing during the robot session (Figure 2A and Figure 2─figure supplement 3B in the revised manuscript) and dHPC firing rates comparably increased in all the nest, proximal and distal foraging regions (Appendix—figure 1B), yet all possible pairs (Robot/nonRobot-nest/proximal/distal) showed spike synchrony and only the Robotpaired distal cells showed decreased spatial correlation between the pre-robot and robot sessions (Figure 2G,H). Therefore, significance of cross-correlation and distal cell remapping cannot be fully attributed to firing increases. In addition, during the pre-surge epoch, BA robot cells did not increase firing rate but still show significant spike synchrony (Figure 1F in the revised manuscript), indicating the probability of correlated firing of the BA and dHPC cells, but not simple firing rate increases in the two regions contributed to significant spike synchrony.

5) Point 3 also applies to the optogenetic stimulation effects on dHPC place cell stability shown in Figure 4. The BA stimulation could be increasing firing rates in some of the dHPC cells resulting in a change in the spatial tuning of the cells.

We found that the optical stimulation of the BA in naïve rats sufficiently and selectively reduced the spatial correlations between pre-stimulation and stimulation sessions only in the distal dHPC cells. Further analyses revealed that optical stimulation of the BA caused increased firing rates in a subset of the distal cells (n=10), but there was no significant difference in spatial correlations between stimulation-neutral and stimulation-excited distal cells (Author response image 2). These results suggest that although BA stimulation elicited firings in some of the dHPC cells, the firing increase did not directly caused changes in spatial tuning of place cells.

**Author response image 2. sa2fig2:** Spatial correlations (pre-stimulation vs. stimulation sessions) of stimulation-neutral distal cells and stimulation-excited distal cells.

6) Another issue is that a majority of the prior literature has not detected any major input from the BA to the dorsal hippocampus but do see strong inputs to the ventral hippocampus. The Petrovich, 2001 and Pitkanen, 2000 papers the authors cite that examined the BA-hippocampus connectivity systematically also don't see a major projection to dCA1, although the Rei paper does suggest that this connection exists. If the direct connections are minor, this doesn't seriously undermine the overall conclusions of the study, but should be acknowledged because the combined effect of the tracing and cross-correlation results in the current version of the paper imply a monosynaptic BA-dHPC connectivity.Suggestions for improvement: The authors should more carefully describe their anatomical findings including from Figure 2-Suppl Figure 2A and show a) broader images in which hippocampal subareas can be seen, b) quantification of the cell number c) think about performing an anterograde tracing experiment looking at terminal innervation of CA1 from BA tracer injections. They could also look at the latency of the cross correlation values in individual cells as well as the latency of the optogenetic stimulation induced increases in firing rate to address this issue. If direct projections are minor, the authors should consider the possibility that the cross-correlations occur through indirect connectivity (which they do to some extent in the Discussion). If so, this doesn't seriously undermine the overall conclusions of the study, but should be acknowledged because the combined effect of the tracing and cross-correlation results in the current version imply a monosynaptic BA-dHPC connectivity.

In our original manuscript, we acknowledged that the amygdala-dCA1 connections are minor (pg. 15): “While the majority of amygdala and dHPC projections is polysynaptic, via dorsal CA3 and ventral HPC areas (McDonald and Mott, 2017; Pikkarainen et al.et al., 1999; Rei et al.et al., 2015), there is also evidence of direct amygdala and dHPC projections (present study; (Petrovich et al.et al., 2001; Pitkanen et al.et al., 2000; Wang and Barbas, 2018); whether this sparse connection contributes to the present results, however, requires further research employing circuit-specific and genetically defined cell type-specific manipulations in transgenic mice models (e.g., selective stimulations of retrogradely labeled BA neurons that directly project to the dHPC place cells).

As for the accuracy of citations, light amygdala-dCA1 projections were revealed via the Phaseolus vulgaris leukoagglutinin tracing method in both Petrovich et al.et al. (2001; see Figures 7, 9, 11) and Pikkarainen et al.et al. (1999; see Figures22B,C) papers. However, both papers emphasized that the amygdala projects primarily to temporal half of the (ventral) hippocampus. Because the functional significance of the sparce amygdala-dCA1 projections is beyond the scope of the present study, we decided to remove the tracing data (see also reviewer 2).

Reviewer #2:In this paper the authors aim at identifying and testing reciprocal influence of the space and threat representations mediated by the dHPC and BA. To do so, they have recorded neuronal activity simultaneously in the dorsal hippocampus (dHPC) and basal amygdala, in a naturalistic foraging task with a looming robot simulating a threatening predator. In this task the animal venture from the nest into a foraging zone to retrieve a food pellet. In one of the sessions, the robot is activated as soon as the animal approaches the pellet, triggering a flight-type reaction (returning to the nest). The authors identify different subsets of BA neurons : threat-responsive and pellet-responsive, and then focus their analysis on threat responsive BA neurons (robot-cells) and study their correlation with the dorsal hippocampus neurons. They then use optogenetics to stimulate BA during foraging, mimicking the BA activation by the robot, and assess the effects of this stimulation on behavior and dHPC place cell stability.The difficulty for assessing place cell activity in threatening situations is that the reaction triggered, in this case hesitation and pausing, constitute strong modification of the exploratory behavior usually requested to analyze place cell activity. It is then hard to pinpoint what is due to actual place-cell property modification and what is due to differences in behavior. The authors have addressed this question in previous studies (not reviewed here). In this paper they reproduce the same results as presented before and add an incremental step by using optogenetics to activate BA and trigger place field instability (they previously established that BA inactivation or lesion prevents place field disruption). A strength of the paper is that optogenetic stimulations of the BA, although triggering the same flight response as the robot, does not seem to trigger the same pausing and hesitations in the approach phase, therefore making this part less prone to spurious results in remapping due to behavioral changes.

The present study is based on simultaneous BA and dHPC recordings whereas our previous study (Kim et al., 2015) was based entirely on sole dHPC recordings. In the original manuscript, we presented the simultaneously recorded BA and dHPC units separately first to classify different BA and dHPC cell types (e.g., robot-responsive, pellet-responsive, nest cells, distal cells, etc.) before revealing their correlated firing characteristics. To accentuate and better explain the novel findings, the revised manuscript now presents (i) the BA and dHPC spike synchrony data first, and (ii) additional data analyses and novel results pertaining to BA and dHPC cell heterogeneity and their dynamic interaction during risky foraging.

We agree with the reviewer that hippocampal place cell firings generally relate to the animal’s movement speed and thus the behavioral changes in our study may contribute to the stability of the place cells.. To examine whether the robot-induced approach/escape behavior or speed changes affected place cell firing, we calculated the relative outward speed between the pre-pellet and pre-surge epochs [(speedpre-surge – speedpre-pellet)/(speedpre-surge + speedpre-pellet)], and the relative inward speed between the post-pellet and post-surge epochs [(speedpost-surge – speedpostpellet)/(speedpost-surge + speedpost-pellet); inbound foraging; Appendix—figure 1F; cf. Kim et al., 2015]. Then, we analyzed the relationship between the relative speed and the spatial correlations (pre-robot vs. robot sessions). Neither the relative outward nor inward speed was correlated with the spatial correlation in both nest/proximal and distal cells, suggesting hippocampal remapping during the robot session cannot be fully explained by the running speed per se. Also, as mentioned in the original manuscript, ‘…the fact that place cell remapping was observed across fear conditioning [i.e., animals displaying freezing] (Moita et al., 2003, 2004), inhibitory avoidance [i.e., animals staying away from the shocked compartment] (Schuette et al., 2020) and ethological fear [i.e., animals fleeing to the nest] (Kim et al., 2015) paradigms, where the animals exhibited dissimilar fear behaviors…’ would indicate that fear rather than the speed of the animal’s movement contributes to alterations in place fields.

As the reviewer acknowledged, we believe our BA-dHPC spike synchrony and optogenetic BA stimulation data corroborate the BA-mediated distal cell remapping during risky foraging in rats.

The major interest and novelty in this study, in my opinion, is the use of this valuable dataset (dHPC and BA simultaneous recordings in a naturalistic foraging task) to look at the fine temporal dynamics between BA and dHPC. Unfortunately, this is also very difficult due to the very small number of correlated cell pairs. Indeed, despite the authors's tracing data and previous literature on it, the presence of direct connections between dHPC and BA is not fully established and, at best, very sparse. This sparsity, or the possibility of correlations through other structures, doesn't make this study less interesting, only more difficult. Indeed, although promising, the author's claims that the BA and HPC reciprocally interact on a short time window around the robot "surge" or closer to the threat is only partially backed by their results that require, at this stage, more careful statistical controls.

As stated in the results, we have simultaneously recorded 1,999 pairs of BA and dHPC cells, of which 714 pairs met the minimum firing rate requirement. From 714 pairs, 30% of pairs (210 pairs) showed significant synchrony to the looming predatory robot. Given the difficulty of simultaneous recordings and a small fraction of fear-responsive amygdalar neurons (e.g., Barot et al., 2009; Gore et al., 2015), we believe our analyses are based on adequate data of correlated cells.

In our original manuscript, we stated that “While the majority of amygdala and dHPC projections is polysynaptic, via dorsal CA3 and ventral HPC areas (McDonald and Mott, 2017; Pikkarainen et al., 1999; Rei et al., 2015), there is also evidence of direct amygdala and dHPC projections (present study; (Petrovich et al., 2001; Pitkanen et al., 2000; Wang and Barbas, 2018); whether this sparse connection contributes to the present results, however, requires further research employing circuitspecific and genetically defined cell type-specific manipulations in transgenic mice models (e.g., selective stimulations of retrogradely labeled BA neurons that directly project to the dHPC place cells).” As our CTB tracing data from two animals were merely to confirm previous reports, as recommended by the reviewer, the CTB tracing data are now removed.

Regarding the CC analysis, we separated pre-surge and post-surge based on our previous study (Kim et al., 2018), where we confirmed that there were behavioral and neural differences between when the animal approached the robot (pre-surge; slower running speed and equivalent proportion of LA and PL leading pairs) and when animals ran away from the robot (post-surge; faster instantaneous running speed and increased proportion of LA leading pairs). In the current study, we also found that different cell pairs showed significant spike synchrony during the pre-surge and postsurge, and their firing patterns were distinct (Figure 1E-H and Figure 1─figure supplement 1C in the revised manuscript).

The testing window (100 ms) for the spike synchrony was set to investigate potential interaction between the dHPC and BA cells, not effects of robot-evoked responses. We chose the 100-ms window based on previous studies showing direct/indirect interaction between two brain structures [BLA-prelimbic cortex, cf. Burgos-Robles et al. (2017); LA-prelimbic cortex, cf. Kim et al. (2018); Lateral septum-hippocampal CA1, cf. Wirtshafter and Wilson (2020)]. For the CC analysis, the firing rate during the 2.5 s epoch (not the overall firing rate) was calculated to determine whether it met the firing rate criterion (i.e., the firing rate must be > 0.1 Hz in both paired cells). Among the total 1999 pairs, 714 pairs met the minimum firing rate requirement, and 30% (210 pairs) of the analyzed pairs showed significant synchrony. To exclude any possibility of covaried or random-overlapped correlations, the original data were analyzed via shift-predictor with ‘100 random shuffles’ and subtracting the shuffled CCs from the raw CCs (Burgos-Robles et al., 2017; Csicsvari et al., 2000; Narayanan and Laubach, 2009). These details and the information of the number of surge events/total recording time are now included in the Materials and methods (Cross-correlation). In addition, narrowing the length of the analysis window (±50 ms) did not change the spatial correlation (pre-robot vs. robot) of the robot cell-paired distal cells (no difference between the spatial correlations of the 100-ms vs. 50-ms cells; Author response image 3), but only reduced the number of distal cells (n=9) synchronized with BA cells, not allowing comparisons between the different types of the distal cells. Given the aforementioned studies and the present data, ±100 ms windows have been remained in our spike synchrony analyses.

**Author response image 3. sa2fig3:** Spatial correlations between the pre-robot and robot sessions from the distal cells that showed significant spike synchrony with the robot cells within ±100 ms or ±50 ms.

– It seems that figures G and H are essentially recapitulating previous work done by the authors (Kim et al.et al. 2015). In particular, how is Figure 1H in the present paper different from figure 2c in the 2015 one?

The present study is based on simultaneous BA and dHPC recordings whereas the 2015 paper was based entirely on sole dHPC recordings. In the original manuscript, we opted to present BA and dHPC recording data separately to first show different BA and dHPC cell types (e.g., robot-responsive, pellet-responsive, nest cells, distal cells, etc.) before presenting their correlated firings. We now realize that this data sequence dampened the novelty of the study and thus we have rearranged the figures to draw attention to the originality of this study, which is that BA and dHPC neural activities were recorded simultaneously and empirically anchored to real dangers that animals face in nature.

– Remapping measures : the peak distance measure isn't clear and is not described in the methods. In the text it is mentioned pauses in the foraging behavior in the robot session. However average running speeds are higher, which I suppose is due to faster running/escaping to the nest. This means there are major speed/behavior differences in the robot and non-robot epochs that could be described/accounted for. In the same line of thought, I suggest an additional criteria for place cells of minimal velocity. This would be ensuring that the remapping is not due to behavioral differences between pre-robot and robot sessions.

How the peak distance was measured is now described in the Materials and methods (Place cell analysis).

To examine whether the robot-induced approach/escape behavior or speed changes affected place cell firing, we calculated the relative outward speed between the pre-pellet and pre-surge epochs [(speedpre-surge – speedpre-pellet)/(speedpre-surge + speedpre-pellet], and the relative inward speed between the post-pellet and post-surge epochs [(speed_post-surge –_ speed_post-pellet_)/(speed_post-surge_ + speed_post-pellet_; Appendix—figure 1F; cf. Kim et al.et al., 2015]. Then, we analyzed the relationship between the relative speed and the spatial correlations (pre-robot vs. robot sessions). Neither the relative outward nor inward speed was correlated with the spatial correlation in both nest/proximal and distal cells, suggesting hippocampal remapping during the robot session cannot be fully explained by the running speed per se. Also, as mentioned in the original manuscript, ‘…the fact that place cell remapping was observed across fear conditioning [i.e., animals displaying freezing] (Moita et al.et al., 2003, 2004), inhibitory avoidance [i.e., animals staying away from the shocked compartment] (Schuette et al.et al., 2020) and ethological fear [i.e., animals fleeing to the nest] (Kim et al.et al., 2015) paradigms, where the animals exhibited dissimilar fear behaviors…’ would indicate that fear rather than the speed of the animal’s movement contributes to alterations in place fields.

– Since the BA is supposed to encode both positive and negative valences, the "pellet-cells" should be included in the analysis, represented as in figure 1D and shown in figure 1F.

The pellet cells were mentioned in the pre-robot session but because they did not respond to the robot, they were categorized into ‘nonRobot’ cells. Also, the BA activity encoding positive valence was not our primary interest and beyond the scope of the study because the pellet success rate during the robot session was too low (< 3%) to analyze positive valence.

– If I understood the methods correctly, the data are normalized separately for the pre-robot and robot-sessions. Is it correct then to substract the z-values (figure 1f) coming from two different normalizations? (+ the number of cells in the text and figure do not match).

Among the robot-responsive BA cells (n=47), only excitatory BA cells (n=45) were included in the analysis for Figure 1F from the original manuscript. Since the Z-value indicates the activity increase over the baseline level in each session, the Z difference between the two sessions “provides a measure of the change in a cell's responsivity (Repa et al.et al., 2001).” However, we have decided to remove the original Figure 1F to highlight the correlational relationship between BA and dHPC pairs, which is the central point of the study.

– I would include a pie chart to clearly state robot, pellet, pellet-robot and neutral cells percentages and numbers in the principal figure.

As suggested, a pie chart is now included in the revised manuscript (*Figure 2B*). As replied in comment #3, the pellet cells were categorized into ‘nonRobot’ cells.

– CTB tracking measures : there is only one example, and the number of animals is not mentioned. There is no quantification. The cell bodies in CA1are blurry and since we don't see the rest of the picture this could well be baseline fluorescence. From the picture if we trust it it also seems that CA1 to BA projecting cells are much more numerous than BA to CA1 projecting cells. This supplementary figure is not convincing to me. I think it should be properly performed or removed. The paper does not need to establish direct connections between the dHPC and the BA to be relevant, provided the results are commented appropriately. Properly establishing this with tracing methods coupled with physiology (ie monosynaptic connections on cross-correlograms) is a whole other endeavor.

As replied above (Reviewer 1, comment #6), our original manuscript stated that the amygdaladCA1 connections are minor (pg. 15; (Petrovich et al.et al., 2001; Pitkanen et al.et al., 2000; Wang and Barbas, 2018). We merely confirmed this using the CTB tracing technique in two animals. As recommended, however, the CTB tracing data are now removed.

– The analyses presented in figure 2 are the one that require more careful control. First, I do not understand why the authors separated pre-surge and post-surge epochs. Then, the 100ms window, although justified by the authors from connectivity data, seem at least partially arbitrary (especially since they also show a long decay -10s- in figure 1F). Finally, restricting the analysis to such little data (the number of surge events/total recording time on which the ccgs are calculated should be stated) induces baselines close to zero and makes it vulnerable to spurious correlations. Then, figures 2C and F illustrate a circular reasoning because the authors are showing significant differences on the averages of cells that were selected based on this difference (during the window).My suggestions are to : – put a limit of a minimal number of spikes in the considered windows (rather than an overall minimal firing rate) – perform the same analysis on randomly chosen 100 windows to evaluate the number of cell pairs that would be identified by chance (or potentially come up with other shuffling strategies). – Vary the length of the chosen window and see if the results are consistent.

First, we separated pre-surge and post-surge based on our previous study (Kim et al.et al., 2018), where we confirmed that there were behavioral and neural differences between when the animal approached the robot (pre-surge; slower running speed and equivalent proportion of LA and PL leading pairs) and when animals ran away from the robot (post-surge; faster instantaneous running speed and increased proportion of LA leading pairs). In the current study, we also found that different cell pairs showed significant spike synchrony during the pre-surge and post-surge, and their firing patterns were distinct (Figure 1E-H and Figure 1─figure supplement 1C in the revised manuscript).

The testing window (100 ms) for the spike synchrony was set to investigate potential interaction between the dHPC and BA cells, not effects of robot-evoked responses. We chose the 100-ms window based on previous studies showing direct/indirect interaction between two brain structures [BLA-prelimbic cortex, Burgos-Robles et al.et al. (2017); LA-prelimbic cortex, Kim et al.et al. (2018); Lateral septum-hippocampal CA1, Wirtshafter and Wilson (2020)]. For the CC analysis, the firing rate during the 2.5 s epoch (not the overall firing rate) was calculated to determine whether it met the firing rate criterion (i.e., the firing rate must be > 0.1 Hz in both paired cells). Among the total 1999 pairs, 714 pairs met the minimum firing rate requirement, and 30% (210 pairs) of the analyzed pairs showed significant synchrony. To exclude any possibility of covaried or random-overlapped correlations, the original data were analyzed via shift-predictor with ‘100 random shuffles’ and subtracting the shuffled CCs from the raw CCs (Burgos-Robles et al.et al., 2017; Csicsvari et al.et al., 2000; Narayanan and Laubach, 2009). These details and the information of the number of surge events/total recording time are now included in the Materials and methods (Cross-correlation). In addition, narrowing the length of the analysis window (±50 ms) did not change the spatial correlation (pre-robot vs. robot) of the robot cell-paired distal cells (no difference between the spatial correlations of the 100-ms vs. 50-ms cells; Author response image 3), but only reduced the number of distal cells (n=9) synchronized with BA cells, not allowing comparisons between the different types of the distal cells. Given the aforementioned studies and the present data, ±100 ms windows have been remained in our spike synchrony analyses.

Reviewer #3:The authors investigate the impact of amygdala activity on coding of place in the hippocampus. They use a paradigm pioneered and established by this group in the past several years, which nicely combines foraging for food with predatory risk/fear. Rats were required to go out of their nest to look for food pellets while in an environment that includes a "scary" robot, designed to mimic predatory risk. This group has previously shown by lesions, inactivation, and disinhibition, that the amygdala regulates risk behavior (fear of the predator-robot) while foraging for food. Additionally, in a follow-up study, they recorded hippocampal place-cells and showed that spatial information is altered ("remapping") by foraging in the robot risky environment and that amygdala lesions prevent this remapping. In the prevent study they seek to establish this framework even further, and they therefore use two strategies: 1. Simultaneous recordings from the amygdala and the hippocampus that allow cross-correlations between single-units in both structures; and 2. Optogenetic excitation of amygdala cells while examining place coding and remapping in the hippocampus. Both approaches provide further evidence to the previous published findings and are overall well performed and reported with clarity. The conclusions are reasonably supported by the data, but the novelty is less clear, and the novel data does not provide clear insights.The strengths of the paper are the simultaneous recordings of spikes in the amygdala and the hippocampus, which allow more direct evaluation of the effect of amygdala activity on remapping of hippocampal place-cells. The main finding is that remapping in the hippocampus occurred mainly at hippocampal cells that showed synchronous activity with robot-responsive amygdala cells. The other strength could be the optogenetic activation of amygdala cells and the effect on behavior and hippocampal remapping, providing a more concrete evidence that it is indeed increased activity in the amygdala that regulates behavior and place coding under threat.The main weakness is that the results do not provide substantial additional insights beyond the already published studies.The cross-correlations are a useful approach, but it is not clear how many of the overall CCs (1999) are independent, namely result from different pairs (vs how many include the same individual neuron). Adding to this the finding that the number of significant CCs is not really high and fluctuates around chance levels (5% or so), makes the finding hard to interpret.

As mentioned above (Reviewer 2, comment #7), the firing rate during the 2.5 s epoch (not the overall firing rate) was calculated to determine whether the CC analysis met the firing rate criterion (i.e., the firing rate must be > 0.1 Hz in both paired cell). Specifically, 714 out of 1999 pairs met the minimum firing rate requirement and 30% (210 out of 714 pairs) of the analyzed pairs showed significant synchrony. To exclude any possibility of covaried or random-overlapped correlations, the original data were analyzed by means of shift-predictor with 100 random shuffles and subtracting the shuffled CCs from the raw CCs (Burgos-Robles et al., 2017; Csicsvari et al., 2000; Narayanan and Laubach, 2009). This information is described in the Materials and methods (Cross-correlation).

Additionally, "non-robot" BA cells are defined as cells that did not respond to any event (robot or pellet), and these are therefore cells which potentially do not have any interest in the paradigm and of lower activity overall. As a result, they do not provide a proper control for CCs that might be related to "remapping" of hippocampal place cells. Finally, the correlation between position along the arena and the "remapping" are very nice but might well be a result of two groups (nest/proximal, and distal), rather than a real gradual correlation. This needs to be analyzed more carefully.

First, the nonRobot cells included the pellet-responsive cells (20.7% of the nonRobot cells) that increased firing to the pellet during the pre-robot session. We agree that the firing rate of the nonRobot cells was lower than that of the Robot cells during the robot session (Author response image 5). However, the peak Z-score of the significant nonRobot cell-pair CCs was comparable with that of Robot cellpair CCs during the pre-surge epoch (Author response image 5, left). Moreover, the peak Z-score of the nonRobot cell-pair CCs was higher than that of the Robot-cell pair CCs during the post-surge (Author response image 4, right). Despite the significant spike synchrony, the nonRobot cell-paired distal place cells exhibited stable place fields across sessions unlike the Robot cell-paired distal place cells (*Figure 2G,H in the revised manuscript*). When further compared with the nest+proximal cells, the distal cells paired with the Robot cells, but not with the nonRobot cells, showed reduction in spatial correlation between the prerobot and robot sessions (Author response image 4). These results indicate that while the nonRobot cells showed correlated firing with the dHPC place cells during the robot encounter, the nonRobot-place cell coupling did not affect the stability of distal place fields.

**Author response image 4. sa2fig4:** Spike synchrony between the dorsal hippocampus and basal amygdala and its impact on spatial correlations of place cells. (**A**) Firing rate differences between Robot, nonRobot and Robot+Pellet cells. (**B**) The dHPC spikes’ Z-score from significant Robot cell, nonRobot cell, and Robot+Pellet cell-paired CCs during the pre-surge (left) and post-surge (right). (**C**) The spatial correlations of nest vs. distal cells that were paired with Robot cells (left) or nonRobot cells (right).

Second, we further examined whether there was a gradual degradation of the spatial correlation as a function of the distance from the safe nest. To do so, we calculated the correlation coefficient between the peak firing location and spatial correlation in (i) nest + distal cells, (ii) proximal + distal cells, (iii) and distal cells only, and (iv) nest + proximal cells. In the first three conditions, we found significant correlations between the X-position and the spatial correlation (nest + distal, *r* = -0.2376; proximal + distal, *r* = -0.4488; distal only, *r* = -0.2459; Author response image 5). There was no such correlation in the nest and proximal cells (*r* = 0.04118; Author response image 5), consistent with our current and previous findings that there were no group differences between the nest and proximal cells (Kim et el., 2015; current study) and animals successfully procured the pellet within the limit of the proximal region (Choi and Kim, 2010). These results confirm that place cells gradually, not area-distinctively, remapped more as their place fields were located farther from the safe nest.

**Author response image 5. sa2fig5:** Spatial correlations in nest + distal cells. (**A**), proximal + distal cells (**B**), distal cells only (**C**), and nest + proximal cells (**D**) between pre-robot and robot sessions are plotted as a function of the peak firing location during the pre-robot session (left, nest; right, end of the foraging apparatus).

The optogenetics is a strength in principle but limited as well. First, if the authors would like to claim that it is indeed BA->HPC, then the power of optogenetics should be used by injecting in one place and stimulating in the other. Second, a control region should be used in addition to the amygdala, to make sure that the stimulation itself (cell excitation) is not in itself a fearful experience for the animal.

We agree with the first comment if BA neurons supposedly influence dHPC neuronal activities via their monosynaptic projections. However, as originally stated in the manuscript, the majority of amygdala and dHPC projections is polysynaptic (via dorsal CA3 and ventral HPC areas). Given the sparsity of direct BA-dHPC connection and other reviewers’ recommendations (i.e., Reviewer 1’s comment #6 and Reviewer 2’s comment #6), the CTB tracing data are now removed.

As for the second comment, we do not have a different brain region stimulation ‘control’ group, which can potentially raise further questions because stimulating a different region can affect/alter other (e.g., nondefensive) behaviors and neural circuits. However, we do have three control conditions, (i) EYFP control (the traditional channelrhodopsin control), (ii) low laser power control, and (iii) optic fiber misplacement control. In all three control conditions, we did not detect defensive behaviors at the timing of light delivery, and in the case of ii and iii control groups, their place cells showed no remapping. These findings suggest that the BA stimulation per se (in the absence of eliciting defensive responses) is not sufficient to cause place cell remapping.

Finally, the "remapping" is the finding that spatial sensitivity has changed, but there is no constructive value in it. In other words, if a place has a predator associated with it, then cells should either represent this location better (to remember and to avoid it), or they should remap to code for something else (e.g. a place that has other pellets and no risk). While one cannot ask the data to show something if it does not, it currently does not provide much beyond the previous studies, and further analyses are required to help understand how remapping is modulated by the amygdala to enable future behavior (e.g. in the post-robot period).

Our remapping data are consistent with the general view that hippocampal remapping ensues as external geometry of the environment or internal states of the animal change (Knierim and McNaughton, 2001; Sanders et al., 2020). We have previously shown that relatively high (but not low) fear state induced hippocampal remapping under the same looming robot situation (Kim et al., 2015). By generating unstable (remapped) place fields near the threat location (where the pellet is situated), the same physical environment might be transiently recognized as a risky context by the animal. Given that hippocampal place cells interacting with fear-coding BA cells selectively showed reduced spatial stability, the fear-induced disruptions of distal place fields may serve to prepare or sensitize the animal to avoid the fearful location. The fact that the firing locations of the distal place cells moved toward the safe nest under a predatory condition (Author response image 6) supports a possible constructive value of the remapping by which place cells instruct animals to “avoid” the location that they (the animals) originally “approached.”

**Author response image 6. sa2fig6:** The comparisons of the peak firing locations of all distal cells during the pre-robot and robot sessions. (**A**) The peak firing locations of all distal cells (n=81) during the pre-robot (top) and robot (bottom) sessions. The x-axis denotes the distance from the nest. (**B**) The distance differences of the peak firing locations between the pre-robot and robot sessions in all distal cells (peak firing location_robot –_ peak firing location_pre-robot_). Below zero indicates that the firing location moved toward the nest.

In addition to what was mentioned in the previous section:1. Some controls are required to make sure that place-coding was not disrupted (remapping) by differences in visiting the distal regions due to the robot risk (different trajectories, less sampling as evident by low success rate etc.).

The original manuscript explicitly addressed this concern (pg. 5): “Because the looming robot prevented the animals from reaching the pellet location, all neural analyses was based on equating the nest-to-foraging distance in each trial of pre-robot, robot and post-robot sessions (Figure 1A).” However, to better clarify this, we added a note “limit for place field analysis” in Figure 1A and now state that “Because the looming robot prevented the animals from reaching the pellet location, the distal cells that had place fields beyond the foraging limit (where the animal did not visit during the robot session) were excluded from place cell analyses to equate the nest-to-foraging distance throughout the sessions.” And as mentioned above, the narrowing of the foraging pathway (21.3 cm width; Figure 1A in the revised manuscript), to ensure that pixel bins were sampled adequately for analyses, minimized the variability in foraging locations and trajectories.

2. The division between proximal to distal should be uniform, either in distance and/or in number of place-cells, both are required for correct interpretation.

In the previous place cell recording study (Kim et al.et al., 2015), we defined the proximal region as the foraging area between 0-25 cm from the nest since rats failed to procure the pellet beyond this proximal-distal boundary when facing a looming robot (Choi and Kim, 2010). We have also reported that when the distal zone was subdivided into two areas (one relatively nearer to the nest and other farther from the nest), there was no reliable difference in spatial correlation and the peak distance between the two distal areas.

3. Remapping (and also BA activity could result from distance from the nest (less safety), and this has been shown in many studies. Currently there is no way to dissociate the effect of the robot from the effect of the distance from the nest (they complement each other). Because all sessions are the same: pre-robot, robot, and post-robot, and there are not specific differences between cells in robot and post-robot, it is hard to dissociate the two factors. A future study can include sessions when a robot surges in a proximal location.

Our spatial correlation data showing distal place fields were stable during the pre-robot and post-robot sessions suggest that remapping cannot be explained merely by the distance from the nest (Figure 2─figure supplement 2C in the revised manuscript). Though we believe the present findings and additional analyses strongly support the idea that robot-induced fear (not the robot itself) instructs different populations of the place cells to encode safe (stable coding) vs. dangerous (unstable coding) locations differently, we also agree with the reviewer that clear dissociation between the effects of the predator and distance from the nest will provide useful information. The reviewer’s excellent suggestion for a future study—to test whether a looming robot closer to the nest would disrupt proximal place fields—is now mentioned in the discussion.

4. Another way to think about the previous concern, is to notice that by definition, a robot cell is a distal cell in the amygdala, and therefore CCs are expected (as nicely exemplified in the scheme in figure 5). a better design can dissociate the two factors.

This also relates to Reviewer 1’s comments #3-5 to which we responded that the significance of CCs cannot be attributed merely to the firing rate changes in both BA and dHPC. To directly compare the peak firing times of the BA Robot cells and dHPC distal cells that showed significant spike synchrony, we aligned the population activities of the Robot-distal pairs to the robot activations (t=0). Robot cell showed robot-evoked firing increases while the distal cells showed double-peaked responses since they fired when the animals visited the distal area, both nearing the robot and escaping from the robot (Figure 2─figure supplement 4A,B in the revised manuscript). Further, the peak firing times of the individual Robot-distal (paired) cells did not fall within a 500-ms time bin except two pairs (marked in red) that showed peaks beyond the post-surge epoch (3.5 s after the robot activation). The firing time differences suggest BA cells tended to respond to robot activation in a time-locked manner whereas dHPC cells fired in a location-specific manner. In addition, 18% (n=37 pairs) of the significant CCs were from dHPCnonRobot cell pairs, indicating firing increase is not always the direct cause of the spike synchrony between BA and dHPC.

5. The findings of CCs, central to the study, are not reported in a clear manner and one has to follow the numbers and the segmentation very carefully.

We now include detailed information concerning CCs in the Materials and methods (Crosscorrelation).

6. It is not clear when exactly photostimulation was delivered during the session.

The photostimulation timing information is now clearly described in the revised manuscript (line 297-298: *approximating the robot trigger distance in Figure 1 experiment, ~ 25 cm from the pellet*).

7. More "constructive" characterization of the remapping would really improve the study and its novelty. Do cells remap to code for a different place? How is it in relation to the pellet locations? Why should place-coding be disrupted if the animal wants to remember the location of a threat ?

Our remapping data are consistent with the general view that hippocampal remapping ensues as external geometry of the environment or internal states of the animal change (Knierim and McNaughton, 2001; Sanders et al.et al., 2020). We have previously shown that relatively high (but not low) fear state induced hippocampal remapping under the same looming robot situation (Kim et al.et al., 2015). By generating unstable (remapped) place fields near the threat location (where the pellet is situated), the same physical environment might be transiently recognized as a risky context by the animal. Given that hippocampal place cells interacting with fear-coding BA cells selectively showed reduced spatial stability, the fear-induced disruptions of distal place fields may serve to prepare or sensitize the animal to avoid the fearful location. The fact that the firing locations of the distal place cells moved toward the safe nest under a predatory condition (Author response image 6) supports a possible constructive value of the remapping by which place cells instruct animals to “avoid” the location that they (the animals) originally “approached.”

[Editors’ note: what follows is the authors’ response to the second round of review.]

Essential revisions:Reviewer #1:The authors have addressed most of my previous concerns and focused the paper around the more novel aspects of the findings related to the role of spiking synchronization between basal amygdala (BA) and dorsal hippocampus (dHPC) neurons during aversive experiences and the role of this process in producing experience dependent remapping of dHPC cells. This remapping is maintained transiently after the predator is removed but while animals are still exhibiting avoidance behaviors and not correlated with changes in firing rate, concerns from past reviews. Notably, BA optogenetic stimulation alone is sufficient to reduce foraging behavior and induce dHPC cell remapping. This provides a physiological mechanism for the authors' previous finding that BA lesions block predator induced dHPC cell remapping.While these results are interesting, there is one remaining issue which should be addressed: Related to Figure 1 showing BA-dHPC synchrony, the authors report the percentage of cells showing synchrony around the pre/post predator encounter and they show that this did not occur in these aversive cell pairs during a non-aversive experiences (i.e. pre/post pellet procurement). However, they did not report any data on the pre/post pellet encounter, explicitly examining the percentage of cells showing significant cross-correlations (CCs) during these periods. This should be included along with an analysis of the overlap between cells with significant CCs during aversive or rewarding experiences if there is a large population of cells showing this for the pellet encounters. This will give the reader a better understanding of the nature of this synchronization, whether it is related specifically to aversive encounters and, if so, whether it occurs in separate populations of cells during these different types of experiences.

This is an excellent suggestion. To determine whether BA and dHPC cells show synchronized spiking during the appetitive experience, we performed additional analyses of CCs during pre-pellet and post-pellet epochs (2.5 s before and after the pellet procurement during the pre-robot session). We found a total of 175 pairs (excluding 4-overlapped pairs in both epochs) that showed significant synchrony during the pre-pellet (*Figure 1─figure supplement 1D* in the updated manuscript) or post-pellet epoch (*Figure 1─figure supplement 1E* in the manuscript). As shown with BA-dHPC cell pairs in robot encounters (pre- and post-surge epochs; Figure 1E and G), the significant (aligned) CCs observed in pellet procurements (pre- or post-pellet epoch) disappeared in other epochs.

We next quantified the selective synchrony in comparison with the pairs from robot encounters (Figure 1─figure supplement 1F in the manuscript). We found that the majority of pairs were selectively synchronized during the pellet or robot experiences (146 pairs during the pellet procurement; 174 pairs during the robot encounter), and there were only 29 pairs that showed significant synchrony during both experiences. The small number of (robot encounter and pellet procurement) overlapped pairs indicates that the BA and dHPC actively communicate not only during aversive but also appetitive experiences by recruiting different sets of BA-dHPC pairs. These results are now mentioned in the Results (Spike synchrony between BA and dHPC units during the predatory encounter).

Reviewer #2:The authors did a fine job addressing most of the concerns, and performed additional analyses producing new figures and results.Yet, I still feel that the main concern: "The main weakness is that the results do not provide substantial additional insights beyond the already published studies.", did not receive a direct and appropriate response in the letter. The CC and the optogenetics are an obvious novelty on one hand, yet on the other, the finding that BA activity in fearful environments is a pre-cursor for remapping of a [any] behavioral correlate that is relevant for the task, is not really novel. If the details and the mechanisms were more convincing, revealing something new about BA-HPC interactions, or about the remapping itself, then it could have been a more interesting study. Overall, I do not have a strong objection to the work, but I admit I am not convinced that it will make a major impact beyond previous work.In addition, some concerns were not fully addressed. For example:The optogenetics is a strength in principle, but limited as well. First, if the authors would like to claim that it is indeed BA->HPC, then the power of optogenetics should be used by injecting in one place and stimulating in the other. Second, a control region should be used in addition to the amygdala, to make sure that the stimulation itself (cell excitation) is not in itself a fearful experience for the animal.Their answer: We agree with the first comment if BA neurons supposedly influence dHPC neuronal activities via their monosynaptic projections. However, as originally stated in the manuscript, the majority of amygdala and dHPC projections is polysynaptic (via dorsal CA3 and ventral HPC areas). Given the sparsity of direct BA-dHPC connection and other reviewers' recommendations (i.e., Reviewer 1's comment #6 and Reviewer 2's comment #6), the CTB tracing data are now removed.This is an argumentative response. If we generalize this type of response, then why do we need any kind of study that injects in one place and stimulate in the other? There are polysynaptic connections between almost any two regions in the brain, hence stimulating one does not demonstrate conclusively that changes that occur in the other are caused by it. Perhaps the BA responses are what they are: BA responses to a fearful environment (as we know fo many years), and HPC remapping occurs due to other inputs from other region independent of the BA? I admit the CCs and the general common sense (hypothesis) in the field makes this less likely, but that is exactly my point about the novelty. Either you show the BA-HPC in a more convincing way, or it remains a correlational nice study with limited novelty.

In the previous version of the manuscript, we acknowledged the limitations of our optogenetic BA stimulation by stating: “The relative contributions of polysynaptic vs. monosynaptic amygdala-dHPC projections to the present results, however, requires further research employing circuit-specific and genetically defined cell type-specific manipulations in transgenic mice models…” We plan to investigate, in the mouse version of ‘approach food-avoid predator’ paradigm, selective stimulations of retrogradely labeled BA neurons that sparsely project to the dHPC place cells or multi-step stimulations/recordings including the di-synaptic BA-CA3/vHPC-dHPC circuits, which is a major research endeavor.

We agree with the reviewer that mechanisms of the causal relationship between the BA-dHPC have not been clearly demonstrated in our study, which is now mentioned in the Discussion. Nonetheless, we believe that this does not diminish the main finding that the place field remapping occurs only in dHPC place cell activities synchronized with BA fear-sensitive cell activities and as a function of escalating risk location. Also, note that the possibility of “HPC remapping occurs due to other inputs from other region independent of the BA” is unlikely given that amygdalar lesions abolished HPC remapping (Kim et al.et al., 2015).

Their answer: As for the second comment, we do not have a different brain region stimulation 'control' group, which can potentially raise further questions because stimulating a different region can affect/alter other (e.g., nondefensive) behaviors and neural circuits. However, we do have three control conditions, (i) EYFP control (the traditional channelrhodopsin control), (ii) low laser power control, and (iii) optic fiber misplacement control. In all three control conditions, we did not detect defensive behaviors at the timing of light delivery, and in the case of ii and iii control groups, their place cells showed no remapping. These findings suggest that the BA stimulation per se (in the absence of eliciting defensive responses) is not sufficient to cause place cell remapping.I do not understand why they did not perform few additional experiments with this control?Their claim that "stimulating a different region can affect/alter other (e.g., nondefensive) behaviors and neural circuits" is directly what they would want to test – that it is indeed BA inputs per-se, and not other changes. Moreover, their response "BA stimulation per se (in the absence of eliciting defensive responses)" indeed raises the possibility that BA stimulation induced a fearful state not because it was in the BA, but because the animal felt something unknown (i and ii do not address this) inducing a fearful state, and hence remapping (and again we are left without knowing if it is via the BA).To emphasize again: I have no doubt that their hypothesis is likely correct, but the study as-is does not tell us much more than we already knew.

We apologize for the oversight of replying ‘BA stimulation per se’ (instead of ‘optical stimulation per se’) when referring to our EYFP and low laser power controls. We now explicitly state the caveats and potential limitations of the single-site stimulation approach in the Discussion. “…while the optogenetic stimulation results may be consistent with the notion that endogenous activation of BA pyramidal neurons disrupted spatial stability of dHPC place cells and impeded successful foraging, a major caveat of our single-site stimulation approach is that neither the possibility of nonspecific stimulation effects nor involvement of other brain regions can be excluded. The latter possibility, however, is unlikely given that amygdalar lesions effectively blocked predatory robot-induced fear and remapping of dHPC place cells (Kim et al.et al., 2015).”

1. Some controls are required to make sure that place-coding was not disrupted (remapping) by differences in visiting the distal regions due to the robot risk (different trajectories, less sampling as evident by low success rate etc.).Their answer: The original manuscript explicitly addressed this concern (pg. 5): "Because the looming robot prevented the animals from reaching the pellet location, all neural analyses was based on equating the nest-to-foraging distance in each trial of pre-robot, robot and post-robot sessions (Figure 1A)." However, to better clarify this, we added a note…This does not answer the main concern. It does address the analyses, but it does not address the potential confound those different trajectories and frequency of visiting distal regions contributed to the remapping. Namely that behaviors that are "outside" of the time/place taken for analyses, took part in remapping. Perhaps I am missing something in their controls?

To clarify, we matched the frequency of visiting the distal zone between the pre-robot and robot sessions (8-10 trials, detailed information can be found in the Materials and methods section; Behavioral paradigms). Specifically, the robot was activated only when the animal visited the pellet vicinity (~ 25 cm from the pellet). In other words, since one robot activation was counted as one robot trial, the animals visited the distal zone the comparable number of times across sessions. Second, selective remapping in distal cells was also observed in our optogenetics experiment, where animals showed relatively linear foraging trajectories and no hesitancy in foraging behavior as there were no discernable external threat (see Figure 4A in the updated manuscript). Hence, the selective remapping in distal cells cannot be accounted by different trajectories or less sampling between pre-robot vs. robot trials.

2. The division between proximal to distal should be uniform, either in distance and/or in number of place-cells, both are required for correct interpretation.In the previous place cell recording study (Kim et al.et al., 2015), we defined the proximal region as the foraging area between 0-25 cm from the nest since rats failed to procure the pellet beyond this proximal-distal boundary when facing a looming robot (Choi and Kim, 2010). We have also reported that when the distal zone was subdivided into two areas (one relatively nearer to the nest and other farther from the nest), there was no reliable difference in spatial correlation and the peak distance between the two distal areas.This is great. Why not repeat the same analyses here and show it is similar? One cannot rely on a previous study for such controls.

As suggested, we have repeated the same analyses (cf. Kim et al.et al., 2015) by separating the distal cells into two groups based on their max firing locations: one with a max firing location nearer to the nest (25-75 cm from the nest, n=53) and the other with a max firing location farther from the nest (75-125 cm from the nest, n=28). We then compared their spatial correlations and peak distances between the pre-robot and robot sessions to confirm our definition of the distal zone. We found no reliable differences in spatial correlations (Author response image 7, Nearer vs. Farther) and peak distances (Author response image 7; Nearer vs. Farther) between the two distal cell sub-groups. These results are consistent with our earlier report (Choi and Kim, 2010; Kim et al.et al., 2015) and suggest that our definition of proximal and distal zones is not arbitrary but supported by behavioral and electrophysiological data. Also, note that remapping observed in both distal cell sub-groups (i.e., Nearer and Farther) addresses, at least partly, the potential confounds of different trajectories affecting remapping in distal cells.

**Author response image 7. sa2fig7:** Spatial correlations and peak distances between the pre-robot vs. robot sessions from the nest, proximal, nearer distal, and farther distal cells. (**A**) Spatial correlations between the pre-robot and robot sessions from the nest, proximal, and distal cells (Nearer distal cells: fired relatively nearer to the nest; Farther distal cells: fired farther from the nest). (**B**) Peak distances between the pre-robot and robot sessions from the nest, proximal, and distal cells (Nearer distal cells: fired relatively nearer to the nest; Farther distal cells: fired farther from the nest).